# Mucosal prime-boost immunization with live murine pneumonia virus-vectored SARS-CoV-2 vaccine is protective in macaques

Jaclyn A. Kaiser[1], Christine E. Nelson [2,12], Xueqiao Liu[1,12], Hong-Su Park [1], Yumiko Matsuoka[1], Cindy Luongo [1], Celia Santos[1], Laura R. H. Ahlers[1], Richard Herbert[3], Ian N. Moore[4,5], Temeri Wilder-Kofie[4,6], Rashida Moore[4,7], April Walker[8], Lijuan Yang[1], Shirin Munir [1], I-Ting Teng[9], Peter D. Kwong [9], Kennichi Dowdell [10], Hanh Nguyen [10], JungHyun Kim[10], Jeffrey I. Cohen[10], Reed F. Johnson[11], Nicole L. Garza[11], Laura E. Via [8], Daniel L. Barber [2], Ursula J. Buchholz [1,13] ✉ & Cyril Le Nouën [1,13] ✉

Immunization via the respiratory route is predicted to increase the effectiveness of a SARS-CoV-2 vaccine. Here, we evaluate the immunogenicity and protective efficacy of one or two doses of a live-attenuated murine pneumonia virus vector expressing SARS-CoV-2 prefusion-stabilized spike protein (MPV/S-2P), delivered intranasally/intratracheally to male rhesus macaques. A single dose of MPV/S-2P is highly immunogenic, and a second dose increases the magnitude and breadth of the mucosal and systemic anti-S antibody responses and increases levels of dimeric anti-S IgA in the airways. MPV/S-2P also induces S-specific CD4[+] and CD8[+] T-cells in the airways that differentiate into large populations of tissue-resident memory cells within a month after the boost. One dose induces substantial protection against SARS-CoV-2 challenge, and two doses of MPV/S-2P are fully protective against SARS-CoV-2 challenge virus replication in the airways. A prime/boost immunization with a mucosally-administered live-attenuated MPV vector could thus be highly effective in preventing SARS-CoV-2 infection and replication.

Since emerging in late 2019, severe acute respiratory syndrome coronavirus 2 (SARS-CoV-2) has caused over 774 million cumulative cases of coronavirus infectious disease (COVID) and over 7 million deaths worldwide[1]. Rapid deployment of SARS-CoV-2 vaccines in late 2020 helped reduce the burden of disease associated with the pandemic. Currently, three SARS-CoV-2 vaccines are available in the United States: two mRNA-based vaccines and a protein subunit vaccine. A fourth vaccine based on a replication-incompetent adenovirus vector was available in the US until May 7, 2023. All of these are injectable vaccines, based on the spike (S) surface glycoprotein. These injectable vaccines stimulate systemic immunity and their use significantly

reduced severe cases of COVID. However, they do not directly induce mucosal immunity in the respiratory tract, limiting their ability to prevent SARS-CoV-2 infection and replication at the primary site of infection[2]. SARS-CoV-2 continues to circulate in communities worldwide, allowing for continuous evolution of new SARS-CoV-2 variants[3]. Mucosal immunization with a live, replicating viral vector would be expected to elicit robust local mucosal immune memory with the ability to effectively restrict SARS-CoV-2 replication at airway surfaces, reducing transmission and the emergence of new variants.

We previously evaluated murine pneumonia virus (MPV) as a live-attenuated vaccine vector for mucosal immunization[4]. MPV is a

pneumovirus that is the murine homolog of human respiratory syncytial virus (RSV). Like RSV, MPV is an enveloped virus with a single-stranded negative-sense RNA genome of approximately 15,000 nucleotides that replicates in the cytoplasm of infected cells. MPV is a promising live vector for intranasal immunization against respiratory viruses because (1) it is highly attenuated in nonhuman primates due to host range restriction, (2) it has a tropism for the respiratory tract, and (3) pre-existing immunity against the MPV vector in humans is low[5]. In a previous study, recombinant MPV expressing the RSV fusion (F) protein, the major RSV neutralization antigen, was highly attenuated for replication in rhesus macaques, but induced high RSV-neutralizing serum antibody titers. Thus, even though its replication in nonhuman primates is low, the MPV vector was strongly immunogenic[4,5].

In the present report, we performed a prime-boost immunization with an MPV vector expressing the prefusion-stabilized version of the SARS-CoV-2 S protein (MPV/S-2P) in rhesus macaques, with a detailed analysis of the local and systemic immune responses and evaluation of protective efficacy.

## Results

### A prime/boost regimen of MPV/S-2P increased the magnitude and breadth of the mucosal anti-S antibody response in the airways, and induced high levels of dimeric anti-S IgA

We previously generated an MPV vaccine vector based on a recombinant version of MPV strain 15 with the L ORF encoding the MPV polymerase partially codon-pair optimized for efficient expression in humans[4,6]. Here, we evaluated a version of this MPV vector expressing the prefusion-stabilized SARS-CoV-2 S protein (S-2P; derived from strain Wuhan-1) in rhesus macaques. To study the safety, immunogenicity, and protective efficacy of one or two doses of this vaccine, delivered by the respiratory mucosal route, we included three groups of macaques (Fig. 1A). Animals received a single intranasal/intratracheal (IN/IT) dose of MPV (empty-vector control, group 1) or MPV/S-2P (groups 2 and 3) on day 0; group 3 received a boost on day 28. After the first dose, shedding of the MPV empty-vector control from the upper airways (UA) was low (<2 $\log_{10}$ PFU/ml, 3/4 animals) and sporadic. Shedding of MPV/S-2P (groups 2 and 3) was detected in the UA of 2/8 animals only, with titers at the limit of detection (Fig. 1B, left panel). In the lower airways (LA), MPV control was detectable in all four animals over 8 days, with peak titers on day 4 post-immunization (pi) (median 4.3 $\log_{10}$ PFU/ml; group 1, $n = 4$). MPV/S-2P was detected in the LA of 3/4 and 2/4 macaques (groups 2 and 3), albeit only through day 6, and titers were significantly lower compared to those of the MPV empty-vector control (Fig. 1B, right panel). After the second dose of MPV/S-2P vaccine (group 3; boost; $n = 4$), shedding was detected only in a single animal in the UA on day 3 at the level of detection of the assay (Fig. 1B). No changes in the macaques' vital signs were observed following IN/IT immunization with MPV or MPV/S-2P (Fig. S1).

A single IN/IT dose of MPV/S-2P efficiently induced mucosal anti-S and anti-receptor binding domain (RBD) IgG and IgA in the UA, detectable in nasal wash (NW) by ELISA (Fig. 2A, B; MPV/S-2P prime; groups 2 and 3 combined, $n = 8$). Surprisingly, despite vaccine replication being undetectable in the UA, the second dose of MPV/S-2P further boosted the mucosal antibody response with a 2.1- to 7.9-fold increase of the anti-S and RBD mucosal IgG and IgA geometric mean titers (GMTs) in the UA from day 25/28 to day 42 (group 3, $n = 4$, 2 weeks post-boost). We also evaluated dimeric IgA because it is highly functional for virus neutralization in the respiratory mucosal environment[7]. While one dose induced dimeric anti-S IgA in two of 8 MPV/S-2P primed animals, the second dose induced dimeric IgA in 3 of 4 animals 2 weeks post-boost (Fig. 2C). In the LA, we detected strong anti-S and anti-RBD IgG and IgA responses after one dose of MPV/S-2P ($n = 8$) that peaked 2 weeks after immunization, with an impressive 50- to 119-fold increases in GMTs ($n = 4$) from day 25/28 to day 37, i.e., 9 days after the second dose ($n = 4$) (Fig. 2D, E). Even though titers

gradually decreased, by 4 weeks after the boost, anti-S IgA and IgG GMTs in the UA and the LA were still above the peak GMTs induced by one dose. In the LA, MPV/S-2P also induced dimeric anti-S IgA, detectable in 6 of 8 animals on day 14 after priming, and in 4 of 4 animals on day 9 post-boost (Fig. 2F).

To evaluate the ability of mucosal antibodies to neutralize the vaccine-matched SARS-CoV-2 strain and variants of concern (VoCs), we used an angiotensin-converting enzyme 2 receptor (ACE2) binding inhibition assay. This assay evaluated the ability of antibodies to inhibit binding of soluble, tagged ACE2 to purified S protein from the vaccine-matched WA1/2020 isolate and nine SARS-CoV-2 variants (Figs. 2G and S2A). In a past study, this assay was more sensitive than SARS-CoV-2 pseudotype neutralization assays[8]. Mucosal antibodies from bronchoalveolar lavage (BAL) collected on day 14 post-priming ($n = 8$) moderately inhibited ACE2 binding to S derived from WA1/2020, Beta (B.1.351) and Delta (B.1.617.2) (median inhibition ranging from 16.6%-54.8%) but not from Omicron variants (Figs. 2G and S2A). However, as early as 9 days post-boost, an increase in ACE2 binding inhibition was detected to S from WA1/2020, Beta, and Delta (90.2%-99.2%), and Omicron sub-lineages (35.2%-60.8%) (Figs. 2G and S2A, $n = 4$). Thus, the second dose of MPV/S-2P induced a remarkable increase in the magnitude and breadth of the mucosal anti-S antibody response.

### A prime/boost regimen of MPV/S-2P increased the magnitude, avidity, and breadth of the anti-S serum antibody response

After a single dose of MPV/S-2P, we detected high titers of serum anti-S and anti-RBD IgA and IgG that remained high through week four after priming (Fig. 3A, B, $n = 8$). The MPV/S-2P boost further increased the peak serum anti-S IgG and IgA GMTs (5.6- and 8.6-fold increase, respectively; $n = 4$), with no significant decrease through week four post-boost. We measured the strength of antibody binding in an ELISA-based avidity assay. The geometric mean of the serum anti-S avidity index increased from 0.87 and 0.94 at 4 weeks after prime to 0.98 and 0.99 at 2 weeks after the boost for IgA and IgG, respectively (Fig. 3A, B), suggesting that further antibody maturation occurred after the boost. Furthermore, the serum anti-S antibodies also conferred antibody-dependent cellular phagocytosis (ADCP) activity after priming, which remained high after the boost (Fig. 3C).

The breadth of the serum-neutralizing antibody response was evaluated in live SARS-CoV-2 neutralization assays against the vaccine-matched strain WA1/2020 or representatives of Alpha or Beta lineages. Neutralizing antibody titers to WA1/2020 were variable and relatively low after the prime [detectable in 6/8 macaques, median 50% SARS-CoV-2 neutralizing antibody titers (ND$_{50}$) of 1.1 $\log_{10}$], with a strong increase (about 9-fold) in 4/4 macaques after the boost (median ND$_{50}$ of 2.0 $\log_{10}$) (Fig. 3D), suggesting that animals were effectively primed for anamnestic B-cell responses despite low or undetectable neutralizing activity after the first dose. The MPV/S-2P boost induced a 13- and 7-fold increase in peak serum neutralizing titers against Alpha and Beta isolates, respectively (Fig. 3D). As expected, no serum anti-S or anti-RBD IgA or IgG or S-specific ADCP activity was induced by the empty vector control (Fig. 3A–C), and all macaques developed high serum neutralizing antibody titers against the MPV vector that remained high in the MPV/S-2P-boosted macaques (Fig. 3D).

To determine whether the mucosal boost increased the breadth of S-specific serum antibodies, we tested the macaque sera for their ability to inhibit the binding of ACE2 to purified S proteins of 22 different SARS-CoV-2 variants (Figs. 3E and S2B). Sera from MPV/S-2P-primed macaques effectively inhibited ACE2 binding to the vaccine-matched S protein of WA1/2020 ($n = 8$, median inhibition of 98.2%), which further increased after boost ($n = 4$; 99.8%). ACE2 binding inhibition to S proteins derived from Alpha, Beta, Delta, and B.1.640.2 (France) lineages was similarly high (medians of 69.5%-88.4%) and further increased post-boost (97.4%-99.4%). ACE2 binding inhibition

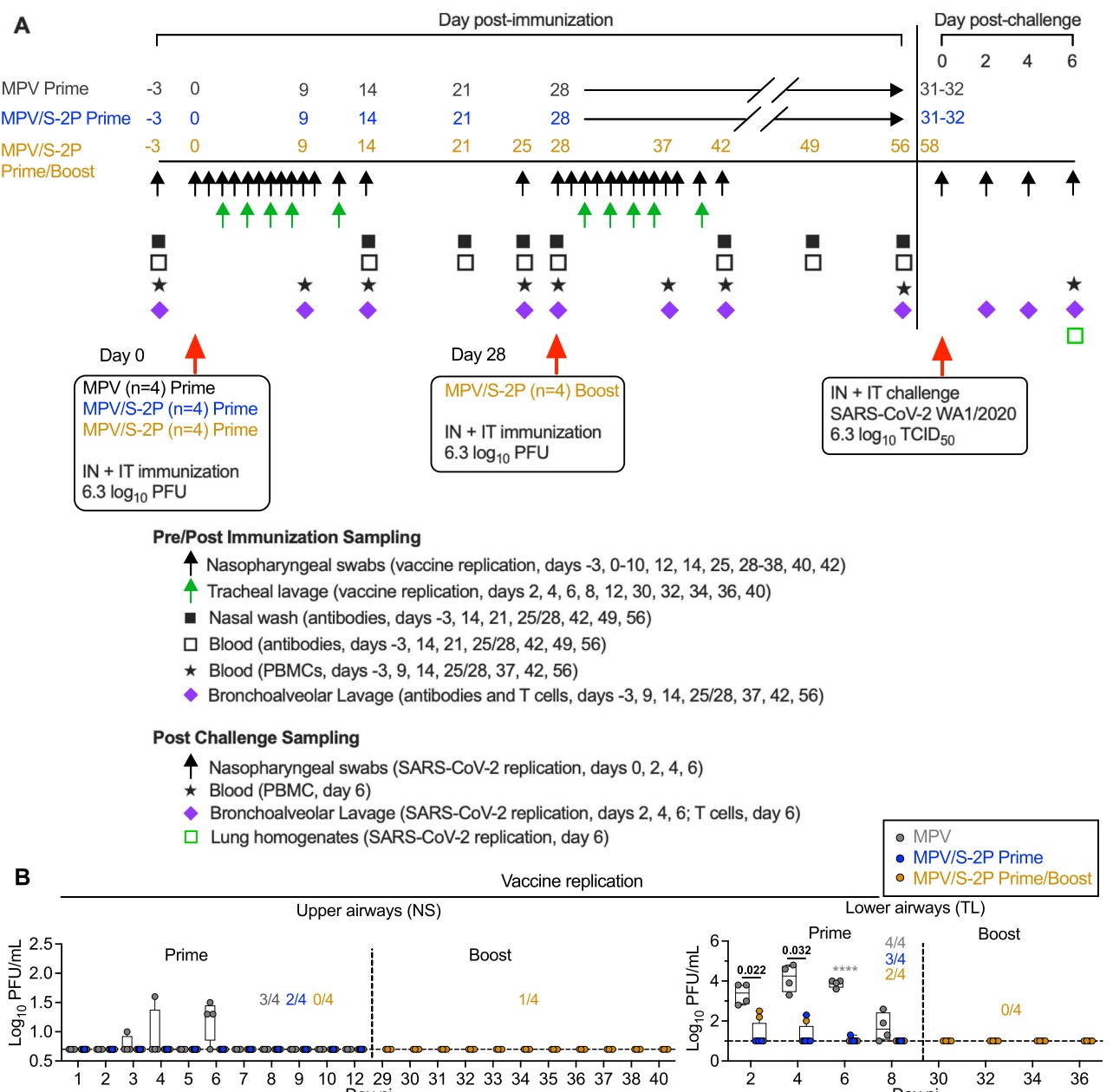

**Fig. 1 | Evaluation of the replication, immunogenicity, and protective efficacy of one or two doses of MPV/S-2P in rhesus macaques. A** Timeline of the evaluation of MPV/S-2P in rhesus macaques. Three groups ($n = 4$ per group) were immunized by the IN/IT route. Group 1 received a single dose of MPV (empty-vector control, gray), and groups 2 (blue) and 3 (gold) received MPV/S-2P. Twenty-eight days after the first dose, macaques of group 3 received a second dose of MPV/S-2P (boost, 6.3 $\log_{10}$ PFU). Thirty-one to 32 days after a single dose (groups 1 and 2) or day 30 after the second dose (group 3, study day 58), macaques were challenged by the IN/IT route with the vaccine-matched WA1/2020 SARS-CoV-2 strain. Vital signs were documented for the duration of the study (Fig. S1). Macaques were euthanized on day 6 post-challenge. **B** Replication of MPV and MPV/S-2P in the upper and lower

airways was evaluated by immunoplaque assay from nasopharyngeal swabs (NS) and tracheal lavage samples (TL), respectively, collected at the indicated days post immunization (pi) from MPV primed ($n = 4$), MPV/S-2P primed ($n = 8$) and boosted ($n = 4$) macaques. Medians (lines), min and max values (whiskers), 25th to 75th quartiles (boxes), and individual values are shown. The limit of detection was 0.7 $\log_{10}$ PFU/ml and 1 $\log_{10}$ PFU/ml for NS and TL, respectively (dotted line). The number of animals per group of macaques with detectable virus replication is indicated. Two-way ANOVA with Sidak post-test; exact $p$ values are indicated for levels of significance $p < 0.05$ unless $p < 0.0001$ (****). Source data are provided in the Source Data file.

to S proteins from Omicron BA.1, BA.2, BA.3, BA.4 and derivatives was variable, ranging from 8.6%–43.2% (Fig. S2B), with marked increases after the boost (53.8%–85.3%). Inhibition of ACE2 binding to S proteins from Omicron BA.5 and derivatives (BF.7, BQ.1, BQ.1.1, and XBB.1) after prime also was modest ($n = 8$, 38.2%–47.2%), and increased after the boost ($n = 4$, 66.5%–77.8%) (Fig. 3E). Thus, S-specific serum antibodies increased over weeks 2–4 post-prime, and further increased post-boost, peaking, for all 4 animals, at 2 weeks post-boost, with high

avidity, breadth and ADCP activity. As expected, no inhibition of ACE2 binding was detected with sera from the four MPV-immunized macaques.

**MPV/S-2P induced S- and RBD-specific peripheral blood B-cell responses**

To extend our analysis of the systemic S-specific immune response to MPV/S-2P, we characterized the kinetics and phenotypes of S-specific

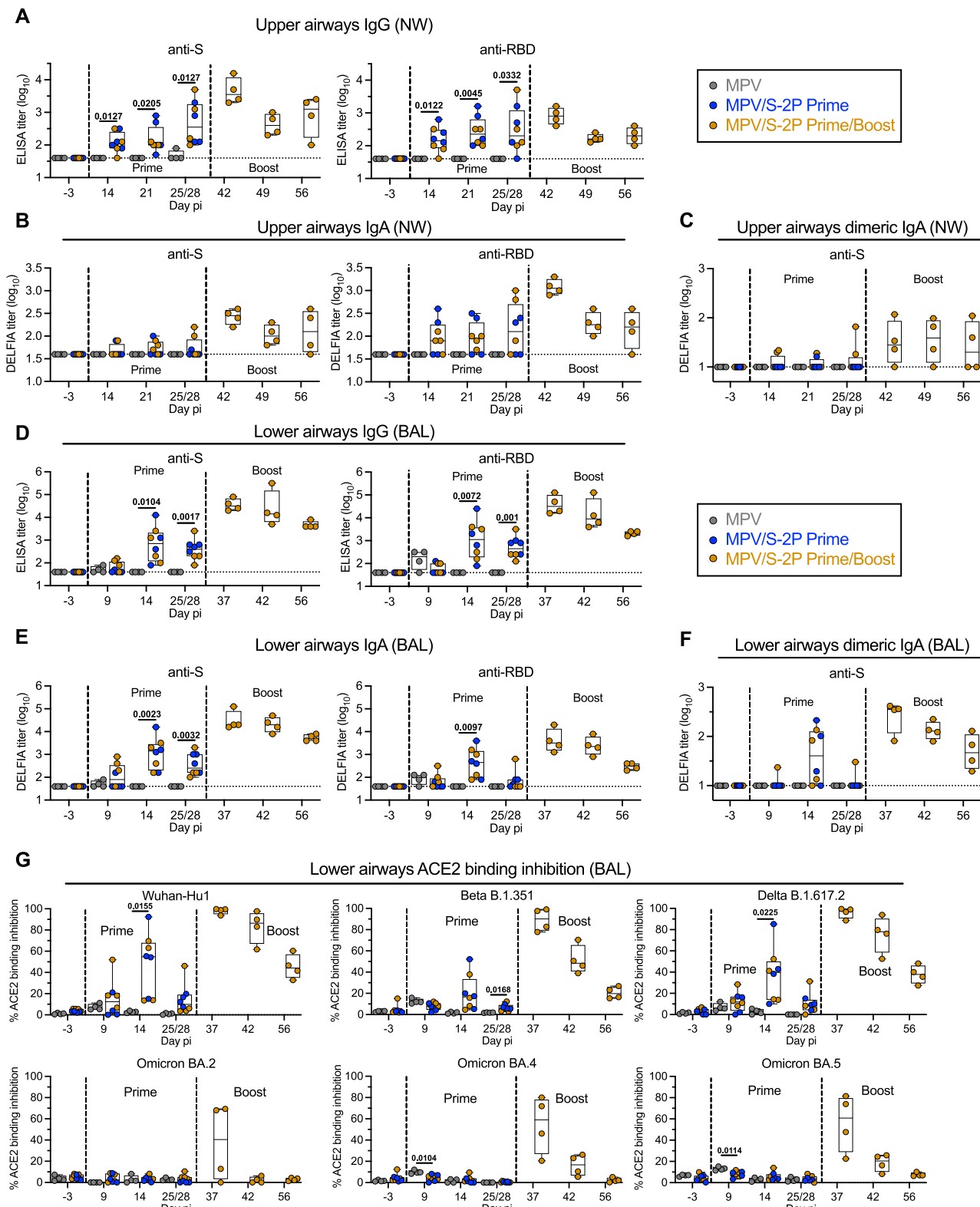

**Fig. 2 | Immunogenicity of MPV/S-2P in the airways.** Immunogenicity in the upper airways (UA) and lower airways (LA) was evaluated from nasal washes (NW) (**A**–**C**) and bronchoalveolar lavages (BAL) (**D**–**G**), collected at the indicated day pi. Spike (S)-specific and receptor binding domain (RBD)-specific IgG (**A**, **D**) and IgA (**B**, **E**) was measured using ELISA and dissociation-enhanced lanthanide fluorescent (DELFIA) assays, respectively (limit of detection: 1.6 log$_{10}$, dotted line). Dimeric anti-S IgA in the UA and LA was also evaluated by DELFIA assay (**C**, **F**; limit of detection is 1.0 log$_{10}$, dotted line). **G** BAL samples were analyzed for their ability to block

binding of tagged, soluble ACE2 to purified S protein from the vaccine-matched SARS-CoV-2 S protein (Wuhan strain) or variants of concern. ACE2 binding inhibition is expressed as % inhibition relative to a no-sample control (see also Fig. S2A). **A**–**G** Medians (lines), min and max values (whiskers), 25th to 75th quartile (boxes), and individual values are shown for MPV primed (*n* = 4), MPV/S-2P primed (*n* = 8) and boosted (*n* = 4) macaques; two-way ANOVA with Sidak post-test; exact *p* values are indicated for levels of significance *p* < 0.05. Source data are provided in the Source Data file.

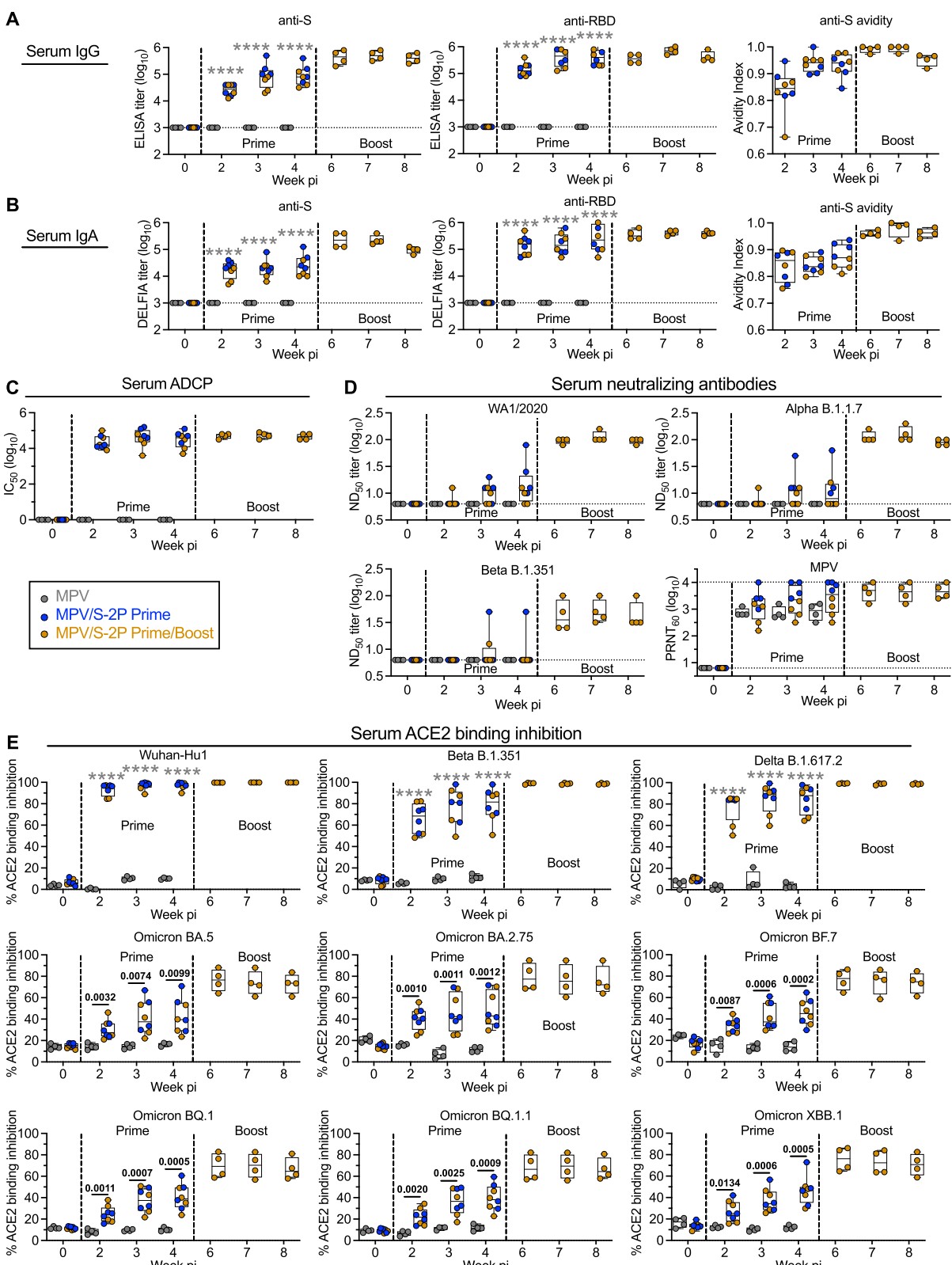

peripheral blood B cells post prime and boost (Fig. 4). Peripheral blood mononuclear cells (PBMCs) were stained with fluorochrome-labeled RBD and S-2P protein probes to identify S-specific B cells that recognize epitopes of the RBD (RBD$^+$/S-2P$^+$), or outside the RBD (RBD$^-$/S-2P$^+$). The cells were further stained using a cocktail of fluorochrome-labeled monoclonal antibodies and analyzed by flow cytometry (Fig. 4A). Analysis was done on live, single, non-naïve (IgD$^-$) CD95$^{+/-}$ B

cells from PBMC collected before immunization and on six time points after prime (groups 2 and 3 combined; $n = 8$) or boost (group 3, $n = 4$; Fig. 1A, see Fig. S3A for the gating strategy).

After MPV/S-2P prime, the frequency of S-specific (RBD$^+$ and RBD$^-$ combined) B cells in the blood peaked at day 14 (median 1.1%), and slightly declined by day 28 (median 0.8%). By day 9 after boost, a rapid recall response was detected, with median frequencies of 1.3%,

**Fig. 3 | Prime/boost regimen of MPV/S-2P induced high level of serum anti-S antibodies with increased avidity, breadth and ADCP activity.** Anti-spike (S) and receptor binding domain (RBD)-specific serum IgG (**A**) and IgA (**B**) measured by ELISA or DELFIA (limit of detection: 3.0 $\log_{10}$, dotted line). Antibody avidity to S, determined by ELISA with or without NaSCN, which strips low-affinity antibodies; avidity index (AI), calculated as ratio of NaSCN-treated vs. PBS-treated serum IgG or IgA titers. **C** Antibody-dependent cellular phagocytosis (ADCP), determined by incubating biotinylated S protein complexed to neutravidin-labeled fluorescent beads with macaque sera, followed by measurement of the phagocytic activity of THP-1 monocytes [ref. 41, modified as described in Supplementary Methods 2]. Data are expressed as the 50% inhibitory concentration ($IC_{50}$) corresponding to the dilution of serum that resulted in 50% of THP-1 cells being FITC positive for binding and/or phagocytosing the S protein. **D** 50% SARS-CoV-2 serum neutralizing-antibody titers ($ND_{50}$) against the WA1/2020, B.1.1.7, and B.1.351 isolates; serum MPV neutralizing-antibody titers, determined by 60% plaque reduction neutralization tests ($PRNT_{60}$). **E** Inhibition of binding of soluble, tagged angiotensin converting enzyme 2 receptor (ACE2) to indicated purified S proteins by serum antibodies, expressed as % inhibition relative to no-serum control (see also Fig. S2B). **A–E** Medians (lines), min and max values (whiskers), 25th to 75th quartile (boxes), and individual values are shown for MPV primed ($n = 4$), MPV/S-2P primed ($n = 8$) and boosted ($n = 4$) macaques; two-way ANOVA with Sidak post-test; exact $p$ values are indicated for levels of significance $p < 0.05$ unless $p < 0.0001$ (****). Source data are provided in the Source Data file.

followed by a steady decrease through day 28 post-boost ($n = 4$, median 0.5% on day 28 post-boost, Fig. 4B). At each time point, RBD⁺-B cells represented approximately one third of S-specific B cells, with the remaining two thirds (RBD⁻/S-2P⁺) exhibiting specificity to undetermined epitopes of S-2P (Fig. 4C).

The majority of S-specific RBD⁺ and RBD⁻ B cells had an IgG isotype (80% median frequency) regardless of the time post-immunization (Fig. 4D, E). The frequency of S-specific IgM B cells was highest at day 14 post-prime (median 12.5% of all S-specific B cells; $n = 8$ macaques) and decreased thereafter, suggestive of isotype switching. S-specific IgA B cells also were detected at each time point. Interestingly, after MPV/S-2P boost, S-specific RBD⁺ IgA B cells (Fig. 4E) were about 4-fold more abundant than RBD⁻ IgA B cells, representing about 16% and 4% of the S-specific B cells, respectively ($n = 4$ animals). About 60% of the S-specific IgG B cells exhibited an activated memory (AM) phenotype (CD21⁻/CD27⁺), with similar proportions for RBD⁺ and RBD⁻ cells (Fig. 4F, G, left). The phenotypes of the S-specific IgA and IgM B cells were more variable, with a lower proportion of activated memory B cells, a greater proportions of cells with tissue-resident like memory phenotype (CD21⁻/CD27⁻), and, especially after the boost, resting memory phenotype (CD21⁺/CD27⁺ or CD21⁺/CD27⁻) (Fig. 4G, center and right). Thus, mucosal immunization with MPV/S-2P induced S-specific B cells in the blood that target the RBD and epitopes on other regions of S, and a recall response after boost of mostly activated memory IgG B cells occurred rapidly.

### Priming with MPV/S-2P by the IN/IT route induces S-specific CD4⁺ and CD8⁺ T-cells in blood and airways that were reactivated by boosting

We also characterized S-specific CD4⁺ and CD8⁺ T-cells in the blood and airways following IN/IT immunization with MPV/S-2P (Fig. 5) by stimulation with pools of overlapping peptides covering the entire SARS-CoV-2 S (PBMC and BAL) or N proteins (control; BAL only). Stimulated cells and unstimulated controls were analyzed by flow cytometry, with gating on live, single, non-naïve (CD95⁺), non-regulatory (FoxP3⁻), CD4⁺ or CD8⁺ T-cells (Fig. S3B). Results from representative macaques are shown in Fig. 5A, D, G, I, while data from all macaques are shown in Fig. 5B, C, E, F, H, J (post-MPV prime: $n = 4$, post-MPV/S-2P prime: $n = 8$; post-MPV/S-2P boost: $n = 4$).

S-specific IFNγ⁺/TNFα⁺ CD4⁺ T-cells in the blood peaked on day 9 after prime (median 0.2%) and declined by day 25/28 (median 0.1%). CD4⁺ T-cells were restimulated 9 days after boost (median 0.2%, Fig. 5A, B) and maintained (median 0.1%) until challenge. The frequency of S-specific IFNγ⁺/TNFα⁺ CD8⁺ T-cells in the blood was variable over the course of the experiment, but all macaques had measurable S-specific IFNγ⁺/TNFα⁺ CD8⁺ T-cells at each day pi (median 0.12–0.21%, Fig. 5A, C).

In the airways, MPV/S-2P induced a greater frequency of S-specific IFNγ⁺/TNFα⁺ CD4⁺ and CD8⁺ T-cells compared to blood (Fig. 5D–F). Airway T-cell responses peaked on day 14 post-prime (medians 8.6% and 7.5% for CD4⁺ and CD8⁺ T-cells, respectively) and decreased by day 25/28 (medians 6.0% and 5.2% for CD4⁺ and CD8⁺ T-cells, respectively).

The MPV/S-2P boost induced a recall response of the IFNγ⁺/TNFα⁺ S-specific T-cells that did not significantly decline between days 9–28 post-boost (medians of 10.5% and 6.0% for CD4⁺ and CD8⁺ T-cells on day 14 post-boost, respectively). As expected, CD4⁺ and CD8⁺ T-cells from BAL did not respond to the pool of N peptides above background (Fig. 5D), and the empty MPV vector did not induce any S-specific T-cells in the blood and airways (Fig. 5A–F).

S-specific IFNγ⁺/TNFα⁺ CD4⁺ and CD8⁺ T-cells in both the blood and BAL of MPV/S-2P-immunized macaques expressed high levels of the proliferation marker Ki-67 on day 9 post-prime that steadily decreased until day 28 post prime (Fig. 5G–J). As expected, Ki-67 expression in S-specific T-cells substantially increased after the boost. Ki-67 expression returned to baseline before challenge (Fig. 5H, J).

### MPV/S-2P immunization induced S-specific T-cells that transitioned to a tissue-resident memory phenotype in airways

We characterized in greater detail the phenotype of the S-specific T-cells in the blood (Fig. S4) and BAL airway samples (Fig. 6). In addition to expressing IFNγ and TNFα, S-specific CD4⁺ T-cells in the airways expressed IL-2 (Fig. 6A, C), consistent with a Th1-biased phenotype. Subsets of these Th1-biased S-specific CD4⁺ T-cells and the large majority of the S-specific IFNγ⁺/TNFα⁺ CD8⁺ T-cells expressed the degranulation markers CD107ab and granzyme B, suggesting that they exhibited cytotoxic activity and were highly functional (Fig. 6A–D). The proportion of polyfunctional S-specific CD4⁺ and CD8⁺ T-cells remained stable after boost (Fig. 6C, D). The functionality of S-specific T-cells in the airways and in the blood was overall comparable (Fig. S4A–D). Interestingly, MPV/S-2P induced a small population of S-specific IL-17⁺ CD4⁺ T-cells in the LA that peaked on day 14 post-prime (median = 0.3%), declined on day 28, and responded to boosting (Fig. 6E, F; median = 0.4% on day 42, i.e., day 14 post-boost).

To confirm that immunization with MPV and MPV/S-2P induced a Th1-biased environment in the airways, we characterized the expression of 36 cytokines in BAL after the priming using a multiplex bead-based immuno-assay (Fig. S5). A transient and moderate increase of the Th1-related cytokines IFNγ, TNFα, and granzyme B was detected on day 9 post-prime with MPV and MPV/S-2P (Fig. S5A); Th2-related cytokines such as IL-4, IL-5, and IL-13 (Fig. S5B) or the Th17-related cytokine IL-17 (Fig. S5C) were not increased, further confirming a Th1-biased response in the airways of macaques. Furthermore, a transient and moderate increase of interferons and chemokines also was measured on day 9 post-prime, but notably, the inflammatory response in the airways of MPV/S-2P-immunized macaques was significantly milder than in MPV-immunized macaques (Fig. S5C).

The phenotype of the S-specific T-cells was further characterized by staining for CD69 and CD103, markers of tissue resident memory (Trm) cells (Fig. 6G–L). On days 9 and 14 post-prime, the largest fraction of the S-specific IFNγ⁺/TNFα⁺ CD4⁺ and CD8⁺ T-cells and IL-17⁺ CD4⁺ T-cells in the airways had a circulating phenotype [48%–71% CD69⁻/CD103⁻ cells (Fig. 6G; gray bars, Fig. 6H, J, L)]. As early as day 37 (9 days post-boost), circulating S-specific T-cells transitioned to Trm

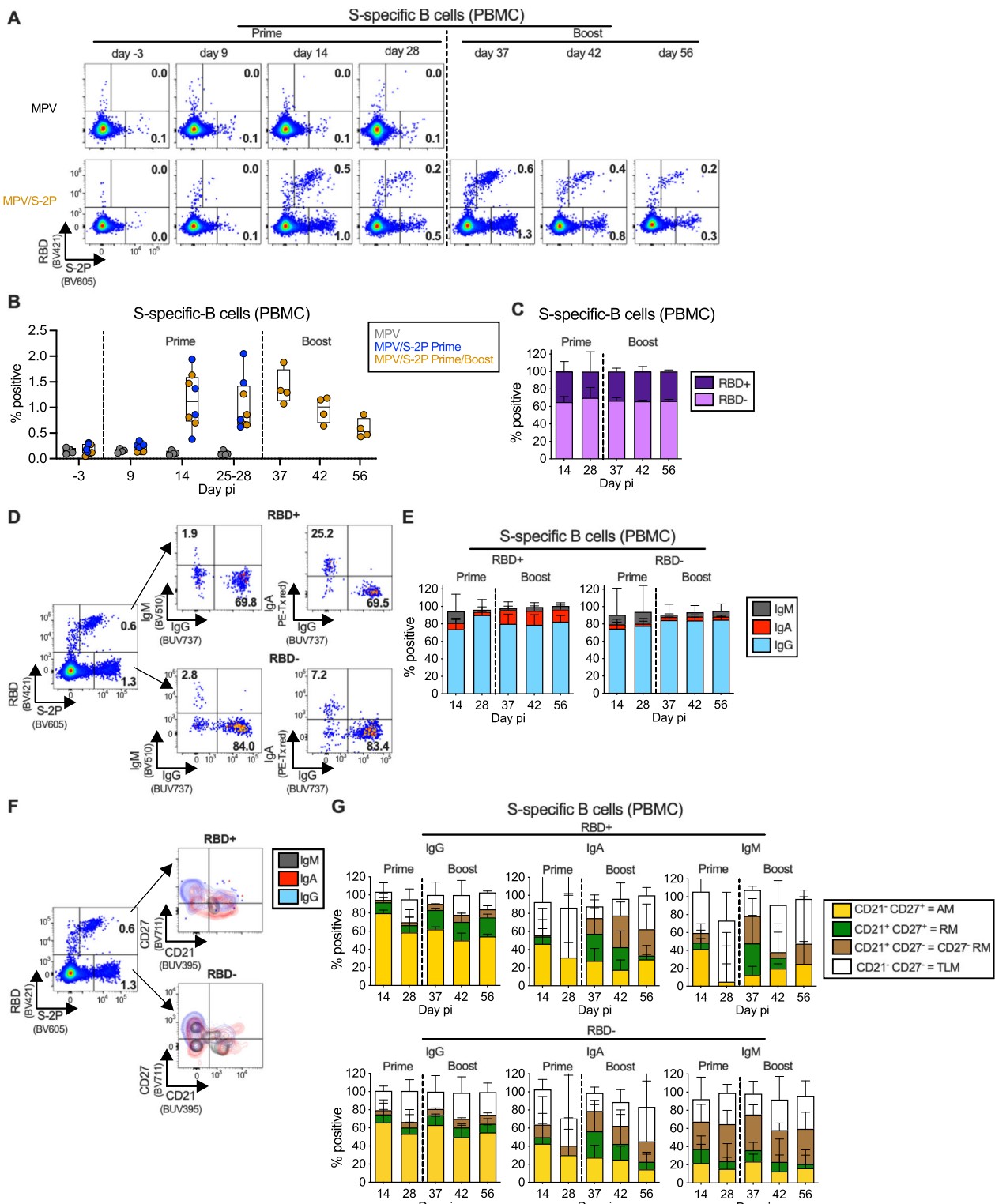

**Fig. 4 | MPV/S-2P immunization induced S-specific B cells in the blood that are restimulated after boosting.** Peripheral blood monocytic cells (PBMC) were stained with fluorochrome-labeled monoclonal antibodies and fluorochrome-labeled SARS-CoV-2 receptor binding domain (RBD) and S-2P probes to identify spike (S)-specific B cells (RBD[+]/S-2P[+] or RBD[−]/S-2P[+], see Fig. S3A for gating strategy). **A, D, F** Dot plots from representative MPV/S-2P- or MPV (A only) immunized macaques showing the frequency of RBD[+] and RBD[−] S-specific B cells. **B** Frequency of S-specific B cells from MPV primed (*n* = 4), MPV/S-2P primed (*n* = 8) and boosted (*n* = 4) macaques [medians (lines), min and max (whiskers), 25th to 75th quartiles (boxes) on indicated days]. **C** Median frequencies with ranges of S-specific B cells in the blood of MPV/S-2P immunized macaques that are RBD[+] (dark purple) or RBD[−]

(light purple) on indicated days (prime *n* = 8; boost *n* = 4 macaques). Isotype class of RBD[+] and RBD[−] S-specific B cells. **D** Representative dot plots from an MPV/S-2P-immunized macaque. **E** Median frequencies with ranges of IgM (gray), IgA (red) and IgG (blue) RBD[+] (left) and RBD[−] (right) S-specific B cells at the indicated day pi (prime *n* = 8; boost *n* = 4 macaques). **F, G** Activated memory (AM; CD21[−]/CD27[+]; yellow), resident memory (RM; CD21[+]/CD27[+]; green), CD27[−] RM (CD21[+]/CD27[−]; brown) or tissue-resident like memory (TLM; CD21[−]/CD27[−]; white) phenotypes of S-specific IgM[+], IgA[+], and IgG[+] B cells of MPV/S-2P-immunized macaques. **G** Median frequencies with ranges (prime *n* = 8; boost *n* = 4 macaques). Source data are provided in the Source Data file.

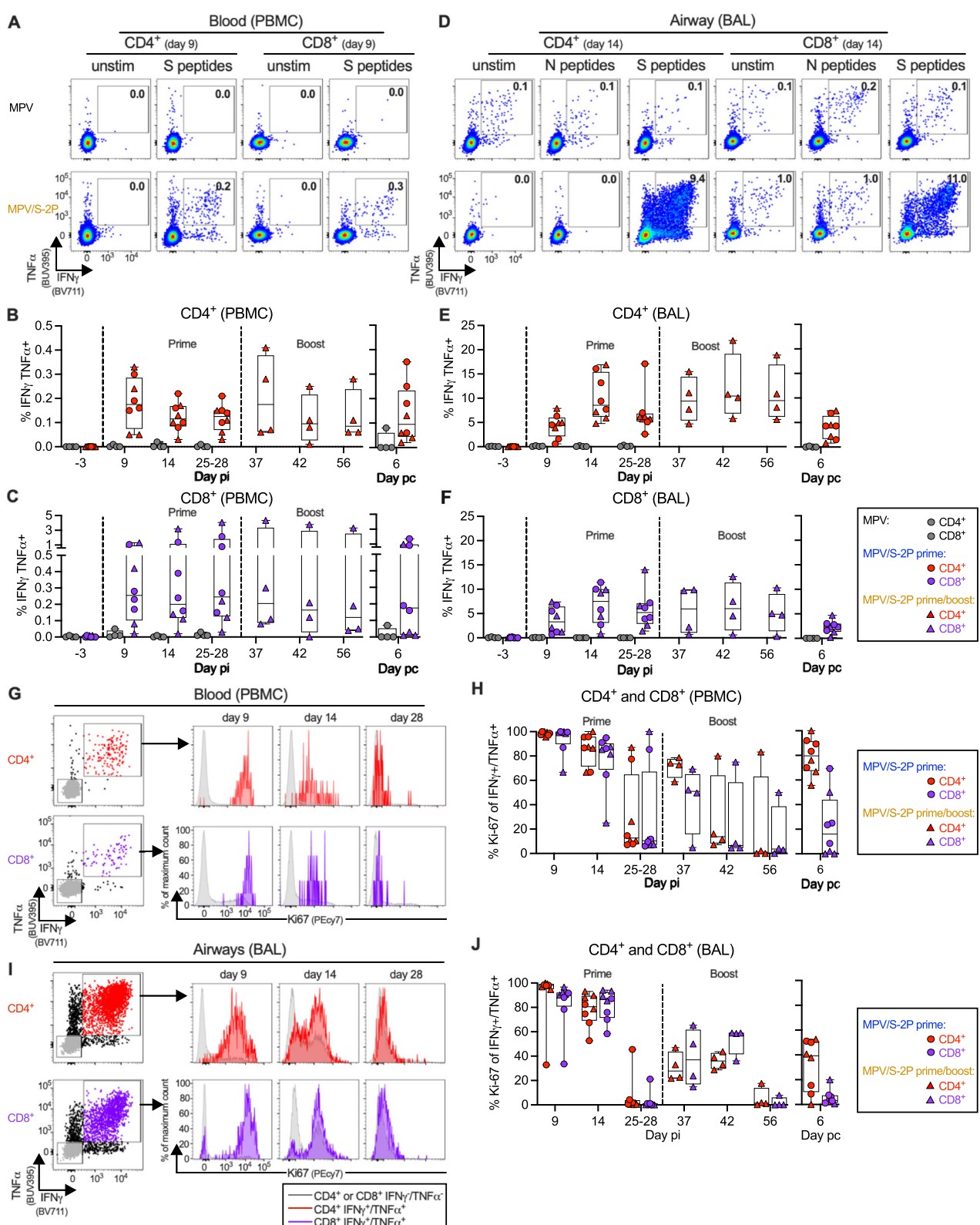

T-cells by the acquisition of CD69 and/or CD103 (blue, green and orange bars), and by day 56 pi (28 days post-boost), 88% and 82% of S-specific IFNγ+/TNFα+ CD4+ and CD8+ T-cells (Fig. 6H, J) and 87% of S-specific IL-17+ CD4+ T-cells (Fig. 6L) exhibited a Trm phenotype. This transition to a Trm phenotype was specific to T-cells from the airways, as S-specific IFNγ+/TNFα+ CD4+ and CD8+ T-cells in the blood remained low for CD69 and CD103 expression (Fig. S4E−H).

**MPV/S-2P immunized macaques were protected from SARS-CoV-2 challenge replication in both upper and lower airways**

To evaluate the protective efficacy of prime and prime/boost regimens with MPV/S-2P, macaques were challenged by the IN/IT route with 6.3 log10 TCID50 of the vaccine-matched WA1/2020 strain of SARS-CoV-2 on days 31 or 32 days after a single dose of MPV or MPV/S-2P ($n = 4$ per group), or on day 58 (30 days after the boost) for the animals that

**Fig. 5 | IN/IT immunization with MPV/S-2P induces S-specific CD4[+] and CD8[+] T-cell responses in the blood and lower airways that are restimulated after boosting.** S-specific CD4[+] and CD8[+] T-cells in blood (**A**–**C**) or bronchoalveolar lavage (BAL) (**D**–**F**). Peripheral blood monocytic cells (PBMC) and bronchoalveolar lavage (BAL) cells from indicated days post-immunization (pi) or post-challenge (pc) were left unstimulated or stimulated with overlapping SARS-CoV-2 spike (S) or (BAL only) nucleoprotein (N) peptides, and processed for flow cytometry (see Fig. S3B for gating). Dot plots showing interferon gamma (IFNγ) and tumor necrosis factor alpha (TNFα) expression by CD4[+] or CD8[+] T-cells from PBMC (**A**) or BAL (**D**) of representative MPV (top) or MPV/S-2P-immunized (bottom) macaques. Background-corrected frequencies of S-specific IFNγ[+]/TNFα[+] CD4[+] (**B**, **E**) or CD8[+] T-cells (**C**, **F**) from PBMC (**B**, **C**) or BAL (**E**, **F**) on indicated days. Expression of proliferation marker Ki-67 by IFNγ[+]/TNFα[+] S-specific CD4[+] (red) or CD8[+] (purple) T-cells from blood (**G**, **H**) or BAL (**I**, **J**) of MPV/S-2P-immunized macaques. Gating and histograms showing evolution of Ki-67 expression by IFNγ[+]/TNFα[+] CD4[+] or CD8[+] S-specific T-cells in blood (**G**) or BAL (**I**) from a representative MPV/S-2P-immunized macaque. IFNγ[−]/TNFα[−] T-cells (gray) are shown for comparison. Frequencies of Ki-67[+] T-cells in IFNγ[+]/TNFα[+] T-cells from blood (**H**) or BAL (**J**). **B**, **C**, **E**, **F**, **H**, **J** Medians (lines), min and max values (whiskers), 25th to 75th quartiles (boxes), and individual values are shown for MPV primed (*n* = 4), MPV/S-2P primed (*n* = 8) and boosted (*n* = 4) macaques. MPV/S-2P-primed and primed/boosted macaques are represented by circles or triangles. Source data are provided in the Source Data file.

received two doses of MPV/S-2P (*n* = 4) (Fig. 1A). Nasal swabs (NS) and BAL were obtained on indicated days post-challenge (pc) to evaluate SARS-CoV-2 replication in the UA and LA. On day 6 pc, animals were euthanized, and lung tissues were obtained.

We quantified challenge virus by RT-qPCR using a SARS-CoV-2 specific TaqMan assay for subgenomic E (sgE) RNA (Fig. 7); sgE mRNA is transcribed in infected cells, not packaged in virions, and serves as a marker for SARS-CoV-2 infection and challenge virus transcription/replication[9,10] (Fig. 7A–C). In the MPV empty-vector control group, we detected sgE mRNA in NS and BAL of 3/4 and 4/4 macaques, with maximal copy numbers on day 2 pc (medians: 3.0 and 4.7 $\log_{10}$ sgE copies/ml in the NS and BAL, respectively; Fig. 7A, B), confirming SARS-CoV-2 challenge virus infection and replication in all empty-vector control animals. sgE RNA remained detectable in the UA of 2 of 4 MPV-control immunized macaques on day 6 pc. However, in MPV/S-2P-primed macaques, only one of four animals had low levels of sgE RNA detectable on day 2 pc in NS only. Notably, in the macaques that had received a prime/boost immunization with MPV/S-2P, no sgE RNA was detected in the NS nor in the BAL, indicating absence of detectable challenge virus replication in the upper and lower airways.

We also evaluated gN RNA in these samples, reflecting the mere presence of challenge virus[10], including residual SARS-CoV-2 inoculum from the high-dose challenge, independent of viral replication (Fig. S6A–C). We detected SARS-CoV-2 gN RNA in NS from all macaques starting on day 2 pi, consistent with the high-dose challenge (Fig. S6A, right panel). On day 2, we also detected high copy numbers of gN in BAL of MPV-control immunized macaques (median of 6.6 $\log_{10}$ copies/ml, Fig. S6), while loads in animals that had received on or two doses of MPV/S-2P were 1000- or 4600-fold lower. These differences are consistent with absence of challenge virus replication in MPV/S-2P immunized animals. Thus, a single dose of MPV/S-2P induced robust protection against SARS-CoV-2 infection in the airways. Based on presence of high levels of gN RNA (i.e., high levels of input challenge virus; Fig. S6) in conjunction with undetectable sgE RNA (i.e., undetectable SARS-CoV-2 transcription, Fig. 7A, B), two doses of MPV/S-2P induced complete protection against challenge virus replication in the airways following high-dose SARS-CoV-2 challenge.

After necropsy, we detected high sgE loads in the lungs of all four MPV empty-vector control-immunized macaques (up to 6.5 $\log_{10}$ sgE copies/g), indicating a high level of SARS-CoV-2 replication in the control group, while all lung tissue specimens from animals that had received one or two doses of MPV/S-2P were negative for sgE mRNA, indicative of the absence of replicating challenge virus and protection against challenge (Fig. 7C). Residual gN RNA was detected in a single sample from the MPV/S-2P one-dose group, while high copy numbers were measured in lung samples from the MPV-control immunized macaques (median of 6.4 $\log_{10}$ copies/g in left middle area of the lungs, Fig. S6C). Thus, SARS-CoV-2 replication was undetectable in the UA and LA of macaques that received two doses of MPV/S-2P, indicating complete protection against challenge.

Finally, we evaluated the CD4[+] and CD8[+] T-cell responses in the blood and airways on day 6 post challenge, focusing on S-specific IFNγ[+]/TNFα[+] cells (Figs. 5B–F and S7). In the blood, we detected a post-challenge increase in S-specific IFNγ[+]/TNFα[+] CD4[+] T-cells in 3/4 and 1/4 MPV/S-2P-primed and -primed/boosted macaques, respectively, with cells from all animals expressing the proliferation marker Ki-67 (Fig. S7C, top). In these animals, there was no detectable increase in S-specific blood CD8[+] T-cells nor an increase in their Ki-67[+] expression (Figs. 5B, C and S6A, C, bottom). In the airways, we did not detect an increase in S-specific IFNγ[+]/TNFα[+] CD4[+] T-cells (Figs. 5E, F and S7B) after challenge, although we detected an increase in their Ki-67 expression in a subset of animals, suggesting CD4[+] T-cell reactivation upon challenge (Fig. S7D, top). There also was no increase in airway S-specific IFNγ[+]/TNFα[+] CD8[+] T-cells, and no increase in their Ki-67 expression (Fig. S7D, bottom).

## Discussion

A SARS-CoV-2 vaccine that can induce immunity directly in the upper and lower respiratory tract—in addition to inducing systemic immunity—should have increased effectiveness because this is the major site of SARS-CoV-2 entry, replication, shedding, and illness[2,11]. Systemic immunity alone has reduced effectiveness in the respiratory tract. For example, studies in humans indicated an approximate 350-fold gradient between serum and mucosal IgG[12]. In addition, induction of local IgA and Trm B and T-cells in the respiratory tract is highly dependent on local immunization[13–17].

In the present study, we used the non-human virus MPV as a live-attenuated viral vector for mucosal delivery to the respiratory tract, designed to express a prefusion-stabilized version of SARS-CoV-2 S. The replication of pneumoviruses is restricted to the respiratory tract, an important safety feature for a vaccine vector. In the present and previous studies[4,5], MPV was strongly attenuated in nonhuman primates, presumably due to a strong host-range restriction. Host-range restriction is typically based on multiple viral genes resulting in high refraction to de-attenuation[18]. Based on their phenotype in primates, MPV vectors are expected to be attenuated but immunogenic in humans. The absence of pre-existing immunity to MPV in humans eliminates concerns of immune restriction of the live-attenuated immunizing vector.

In a previous study, we found that a single IN/IT immunization of macaques with MPV/S-2P induced high levels of serum anti-S and anti-RBD IgG that were comparable to levels measured in the plasma of COVID-19 convalescent individuals[6]. However, in a SARS-CoV-2 neutralization assay, serum neutralizing antibodies titers were variable among MPV/S-2P-immunized macaques. This prompted us to evaluate if a prime/boost regimen of MPV/S-2P would improve the S-specific immune response in SARS-CoV-2 naïve animals. Indeed, our results show that despite the absence of detectable replication in the UA and LA in 3 of 4 animals, presumably due to inhibition by anti-vector immunity induced by the first dose, the MPV/S-2P boost strongly increased the magnitude of the mucosal anti-S IgG and IgA antibody response and induced high levels of S-specific dimeric IgA in the UA and LA. Dimeric IgA is highly neutralizing and non-inflammatory[7]. Nasal dimeric IgA but not IgG has been shown to correlate with nasal

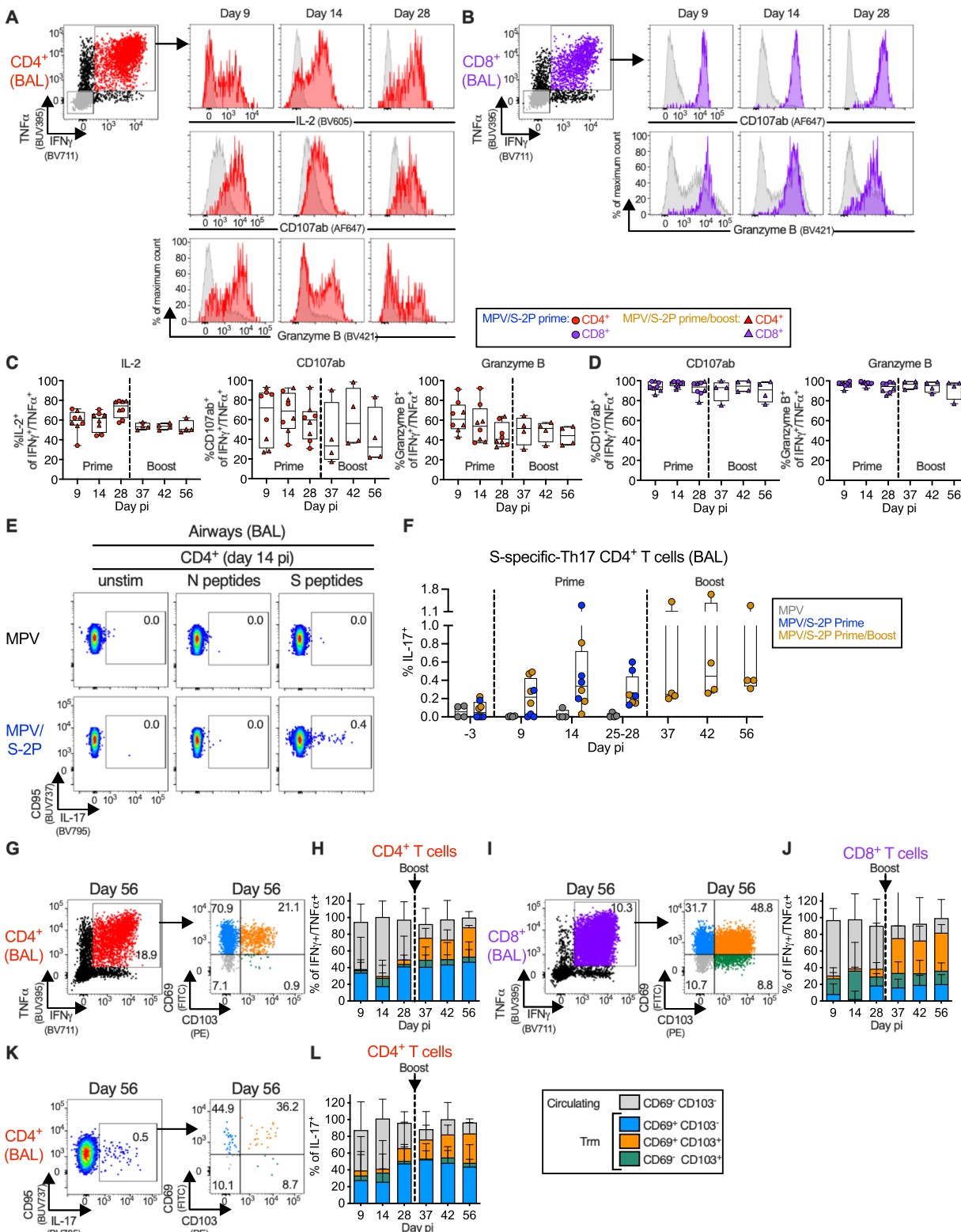

neutralization of SARS-CoV-2[19], or, in controlled human infection models, with protection against RSV[20]. The MPV/S-2P prime/boost regimen also strongly increased the magnitude, breadth, and avidity of serum anti-S antibodies, suggesting further affinity maturation following the boost. Serum anti-S antibodies also displayed Fc receptor-mediated ADCP activity that increased following boost, enhancing the functionality of MPV/S-2P-induced serum antibodies and their breadth against SARS-CoV-2 VoCs[21].

As expected, the second dose resulted in a quick recall response of mostly activated peripheral blood memory IgG B cells. Approximately one-third of the S-specific B cells recognized epitopes of the RBD which represents approximately 20% of the S protein. Thus, the B-cell response was modestly biased toward the RBD, consistent with previous studies showing that RBD was the primary antibody target for neutralizing antibodies[22,23]. RBD-specific IgA antibodies in particular have been associated with SARS-CoV-2 protective immunity[24].

**Fig. 6 | MPV/S-2P induced S-specific Th1 and Th17 CD4⁺ and granzyme B expressing CD8⁺ T-cells in lower airways that remained functional after boost and transitioned to tissue-resident memory phenotype. A–L** Bronchoalveolar lavage (BAL)-derived T-cells were stimulated with overlapping SARS-CoV-2 spike (S) or nucleoprotein (N) peptides or kept unstimulated and processed for flow cytometry. Dot plots, histograms (**A, B**), and frequencies (**C, D**) of S-specific interferon gamma/tumor necrosis factor alpha (IFNγ⁺/TNFα⁺) CD4⁺ (**A, C**, red), CD8⁺ (**B, D**, purple), and IFNγ⁻/TNFα⁻ (**A, B** only, gray) T-cells expressing interleukin 2 (IL-2) (CD4⁺ T-cells only), CD107ab and granzyme B in BAL from MPV/S-2P-primed (**C, D**, circles, $n = 8$) and -boosted (**C, D**, triangles, $n = 4$) macaques on indicated days (**C, D**, medians (lines), min and max values (whiskers), 25th to 75th quartiles (boxes), and individual values are shown; see Fig. S3B for full gating). **E, F** MPV/S-2P induced a small population of S-specific T helper 17 (Th-17) CD4⁺

T-cells in airways that are restimulated after boost. **E** Dot plots from representative macaques and (**F**) frequencies of S-specific CD95/IL-17⁺ CD4⁺ T-cells from BAL on indicated days; medians (lines), min and max values (whiskers), 25th to 75th quartiles (boxes), and individual values are shown for MPV primed ($n = 4$), MPV/S-2P primed ($n = 8$) and boosted ($n = 4$) macaques. **G–L** S-specific T-cells from BAL transition to circulating (CD69⁻/CD103⁻, gray) and tissue-resident memory [Trm; CD69⁺/CD103⁻ (blue), CD69⁺/CD103⁺ (orange), CD69⁻/CD103⁺ (green)] phenotypes. Dot plots (**G, I, K**) and frequencies (**H, J, L**) of Trm phenotypes in S-specific IFNγ⁺/TNFα⁺ CD4⁺ (**G, H**), CD8⁺ (**I, J**) and CD95⁺/IL-17⁺ CD4⁺ T-cells (**K, L**) from BAL on indicated days. **H, J, L** Median frequencies, stacked, with ranges, from MPV/S-2P primed ($n = 8$) and boosted ($n = 4$) macaques. Figure S4 shows Trm markers in PBMC-derived T-cells. Source data are provided in the Source Data file.

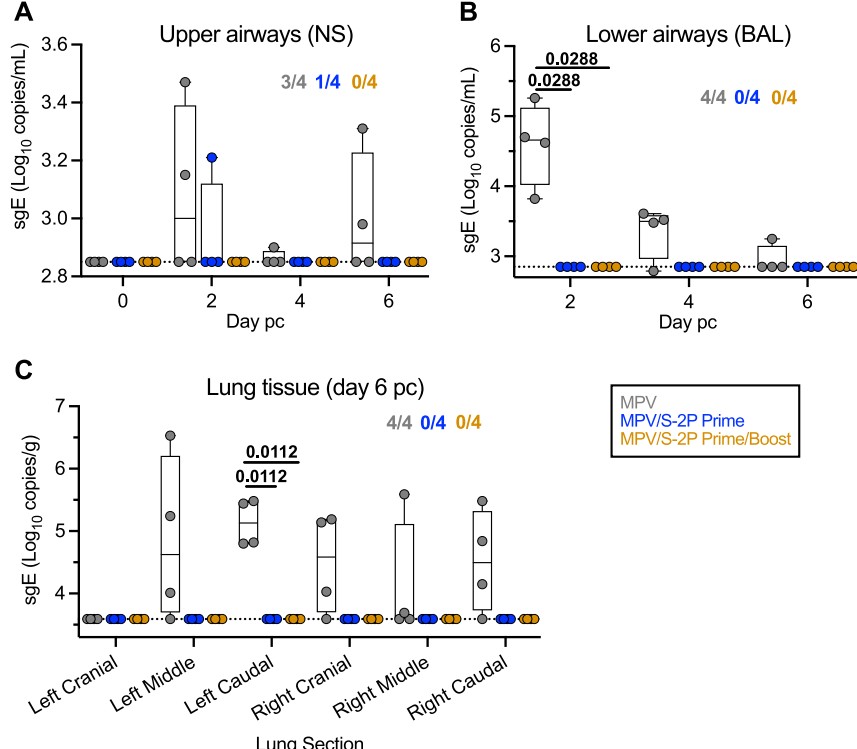

**Fig. 7 | Efficacy of one or two doses of MPV/S-2P against SARS-CoV-2 challenge virus replication.** On days 31 or 32 post-prime [MPV- (gray) and MPV/S-2P-immunized macaques, blue] or on day 58 (corresponding to day 30 post-boost; MPV/S-2P-immunized macaques, gold) ($n = 4$ animals per group), macaques were challenged IN/IT with 6.3 log₁₀ TCID₅₀ of SARS-CoV-2, strain WA1/2020 (Fig. 1A). SARS-CoV-2 subgenomic E (sgE) RNA in the upper (**A**) and lower airways (**B**) following challenge. Nasopharyngeal swabs (NS) and bronchoalveolar lavages (BAL) were collected, and SARS-CoV-2 sgE mRNA, indicative of transcription/de-novo RNA

synthesis and challenge virus replication, was quantified by RT-qPCR. **C** SARS-CoV-2 replication in lung tissues. On day 6 pc, SARS-CoV-2 sgE was quantified by RT-qPCR in six regions of lung tissues from each macaque. In each graph, the number of macaques with detectable sgE is indicated. Limit of detection: 2.85 log₁₀ copies/ml for NS and BAL; 3.6 log₁₀ copies/g for lung tissue. **A–C** Medians (lines), min and max values (whiskers), 25th to 75th quartiles (boxes), and individual values are shown two-way ANOVA with Sidak post-test; exact $p$ values are indicated for levels of significance $p < 0.05$. Source data are provided in the Source Data file.

Interestingly, following the MPV/S-2P boost, RBD⁺ IgA B cells were more abundant than RBD⁻ IgA B cells.

The systemic and mucosal IgA recall response after the second dose of MPV/S-2P in macaques was substantial. In a recent study following nasal and plasma antibody responses 1 year after COVID-19 hospitalization, individuals who later received an injectable SARS-CoV-2 vaccine mounted systemic IgG and IgA recall responses and nasal IgG responses, but the nasal IgA responses were negligible[13]. This suggested that the nasal IgA responses are compartmentalized from the systemic responses[13,25]. Based on the strong mucosal recall responses that were induced by the second dose of MPV/S-2P in our study, we would predict that an intranasal vectored vaccine will induce a mucosal IgA recall response in the respiratory tract in individuals with prior SARS-CoV-2 infection, improving protection

against re-infection and replication at the mucosal entry sites of SARS-CoV-2.

MPV/S-2P induced S-specific peripheral blood CD4⁺ and CD8⁺ T-cells with frequencies comparable to other vaccine platforms[26,27], but their frequencies in the airways largely exceeded what has been described for mRNA vaccines[28] and were comparable to those induced by mucosal immunization with the parainfluenza virus vector B/HPIV3/S-6P[8]. A large proportion of these highly functional S-specific CD4⁺ T-cells expressed Th1-related cytokines. Interestingly, a small population of S-specific CD4⁺ T-cells expressed IL-17 which has also been identified following immunization with the B/HPIV3/S-6P vector[8] and thus seems to be induced by live-attenuated viral vectors. Such Th-17 cells were previously shown to facilitate the recruitment of T and B cells in the lung, resulting in a faster IgA recall response[29]. The

S-specific CD4[+] and CD8[+] T-cells in the blood and airways appeared to also be highly functional, expressing high levels of granzyme B. These S-specific CD4[+] and CD8[+] T-cells were restimulated after boost while they transitioned to a Trm phenotype, providing a critical line of defense for the control of respiratory infections[14–16]. In convalescent COVID patients, Trm CD8[+] T-cells were detected in the lungs up to 10 months after initial infection, likely contributing to long-term protection against SARS-CoV-2[17].

Previous studies evaluated intranasal immunization with non-replicating adenovirus-based COVID-19 vaccine candidates in Phase 1 studies[30] or in a heterologous prime/boost study using the antigenically-distinct candidates Sad23L and Ad49L in nonhuman primates[31]. These candidates were generally well-tolerated, but following delivery by the intranasal route, their immunogenicity was relatively low[30,31]. Another Phase 1 study evaluated the immunogenicity of a Newcastle disease virus vector expressing the SARS-CoV-2 S protein, delivered intramuscularly or intranasally. This vaccine induced S-specific antibody and T-cell responses, but the immunogenicity following intranasal delivery also was relatively weak[25]. Intranasal delivery of subunit and mRNA vaccines is also being evaluated, but inactivated or non-replicating vaccines may require adjuvants for an efficient induction of a mucosal T-cell response, adding layers of complexity to the vaccine development path[30].

At the time of challenge in this study (4 weeks post-prime or 4 weeks post-boost), titers of S-specific serum antibodies were high, while mucosal antibodies had declined from their peak titers at 2 weeks post-immunization, and the majority of antigen-specific CD4[+] and CD8[+] T-cells had differentiated into Trm cells. This combination of systemic and mucosal immunity was strongly protective against SARS-CoV-2 challenge. Specifically, only one animal had traces of challenge virus replication detectable in the upper airways (measured as sgE RNA) after a single dose of MPV/S-2P despite high levels of challenge virus (measured as gN RNA). Protection increased after the second dose, as no animals had measurable challenge virus replication. While strong protection in the lower airways has also been shown for injectable vaccines, including mRNA and adenovirus vectored vaccines[32–34], protection by MPV/S-2P against SARS-CoV-2 challenge virus replication in the upper respiratory tract was more robust compared to injectable vaccines and predicts that MPV/S-2P immunization may restrict SARS-CoV-2 transmission in immunized individuals.

In summary, a single dose of MPV/S-2P elicited strong systemic and mucosal immunity (antibodies, B cells, and T cells) in macaques. The second intranasal dose increased the magnitude, avidity, ADCP, and breadth of the mucosal and systemic anti-S IgG and IgA antibody responses, and induced restimulation and proliferation of S-specific T-cells. One or two doses were protective against SARS-CoV-2 challenge. Even though we detected anti-MPV vector antibodies following the first dose of MPV/S-2P, the increase in anti-S immune responses following a second dose was remarkable. Future studies will evaluate the durability of immunity, and we will evaluate MPV/S-2P in homologous and heterologous prime/boost studies with mRNA vaccines or other intranasal vaccine candidates, e.g., the pediatric B/HPIV3/S-6P live-attenuated vector described previously[8]. B/HPIV3/S-6P was primarily designed as a pediatric vaccine to protect against HPIV3 and COVID; with MPV/S-2P, a live intranasal vaccine candidate without pre-existing immunity in humans is available for clinical evaluation in adults. A Phase 1 study to evaluate the safety and immunogenicity of intranasal immunization with MPV/S-2P is in preparation. MPV vaccine candidates expressing spike proteins from currently circulating SARS-CoV-2 variants are in manufacture to prepare for evaluation beyond Phase 1.

## Methods
### Study design
The objectives of this study were to characterize in rhesus macaques the safety, immunogenicity, and protective efficacy of one or two

mucosal doses of the live MPV-vectored SARS-CoV-2 vaccine MPV/S-2P. We evaluated vaccine shedding, mucosal and systemic antibody and T-cell responses and B cell responses following mucosal administration of one or two doses of MPV/S-2P or empty-vector control. We also evaluated immune responses and protection following SARS-CoV-2 challenge. We used four animals per group, consistent with prior studies of mucosal vaccines in nonhuman primates. Results from animals primed with MPV/S-2P were combined for some analyses. The sample size for each group is indicated in the figure legends. Animals were randomly assigned to experimental groups. The study was performed unblinded.

### Cell lines
African green monkey kidney Vero (ATCC CCL-81) and Vero E6 (ATCC CRL-1586) cells were grown in OptiMEM (Thermo Fisher) supplemented with 5% FBS. Vero cells were used to expand MPV and MPV/S-2P virus stocks. Vero E6 cells were used for SARS-CoV-2 neutralization assays and titrations. SARS-CoV-2 virus stocks were expanded on Vero E6 cells or Vero E6 cells stably expressing human TMPRSS2[35].

### SARS-CoV-2 virus stocks
The SARS-CoV-2 USA-WA1/2020 challenge virus (lineage A; GenBank MN985325 and GISAID accession ID: EPI_ISL_404895; obtained from Natalie Thornburg, Sue Gerber, and Sue Tong, Centers for Disease Control and Prevention [CDC], Atlanta, GA) was passaged 6 times on Vero E6 cells stably expressing human TMPRSS2. The USA/CA_CDC_5574/2020 isolate (lineage B.1.1.7, GISAID: EPI_ISL_751801; CDC) and the USA/MD-HP01542/2021 isolate (lineage B.1.351, GISAID: EPI_ISL_890360; sequence deposited by Christopher Paul Morris, Chun Huai Luo, Adannaya Amadi, Matthew Schwartz, Nicholas Gallagher, and Heba H. Mostafa, The Johns Hopkins University; isolate obtained from Andrew Pekosz, The Johns Hopkins University, Baltimore, MD) were passaged on Vero E6 cells stably expressing TMPRSS2. Titration of SARS-CoV-2 stocks was performed by determination of the TCID$_{50}$ in Vero E6 cells. Whole-genome Illumina deep sequencing analysis confirmed that the complete genome sequences of the SARS-CoV-2 used for experiments were identical to that of consensus sequences, except for minor backgrounds of reads (<10%). All experiments with SARS-CoV-2 were conducted in Biosafety Level-3 containment laboratories approved by the US Department of Agriculture and CDC.

### Generation of MPV and MPV/S-2P
The recombinant MPV vector used in this study was based on a previously-described reverse-genetics system for MPV strain 15 (GenBank AY729016)[5]. In this version, the downstream 67% of the L ORF were codon-pair optimized for efficient expression in humans. MPV/S-2P[6] contains an additional gene encoding the full-length ORF encoding the 1273 amino acid S protein derived from the ancestral Wuhan-Hu-1 sequence (GenBank MN908947), codon optimized for human expression, with two prefusion-stabilizing proline substitutions (aa 986 and 987), and four aa mutations (RRAR to GSAS, aa 682–685) to ablate the furin cleavage site between S1 and S2[36]. Virus stocks were amplified in Vero cells, and the genomic sequences were confirmed by Sanger sequencing of overlapping RT-PCR amplicons.

### Immunization and challenge of rhesus macaques
The study was approved by the NIAID Animal Care and Use Committee and included twelve juvenile and young adult (31–59 months of age) male Indian-origin rhesus macaques (*Macaca mulatta*), seronegative for SARS-CoV-2. The study was not powered to consider sex as a variable. Animals were immunized intranasally (0.5 ml per nostril) and intratracheally (1 ml) with a total dose of 6.3 log$_{10}$ PFU of MPV empty vector or MPV/S-2P. On day 28, 4 of the 12 animals received a second dose of MPV/S-2P. Animals were monitored daily from day −3 through the end of the study.

The schedule for the macaque experiment is shown in Fig. 1A. On indicated days, nasopharyngeal swabs (NS) and tracheal lavages (TL) were performed to evaluate vaccine virus replication. NS were collected using cotton-tipped applicators, placed in 2 ml Leibovitz (L-15) medium with 1x sucrose phosphate (SP) as stabilizer and vortexed for 10 s. TL were mixed 1:1 with L-15 medium containing 2x SP. Nasal washes (NW) for analysis of mucosal antibodies in the UA were performed on indicated days (Fig. 1A) using 1 ml of Lactated Ringer's solution per nostril. Aliquots of all samples were snap-frozen in dry ice and stored at −80 °C. Blood was collected on days −3, 9, 14, 21, 25 or 28 after the prime and on days 37, 42, 49 and 56 from the macaques that received the MPV/S-2P boost dose for analysis of serum antibodies and/or peripheral blood mononuclear cells (PBMCs). Bronchoalveolar lavages (BAL) were performed for analysis of mucosal antibodies and mononuclear cells in the LA.

Four weeks after immunization (or 4 weeks after boost for animals that received a prime/boost regimen), macaques were transferred to an Animal Biosafety Level 3 (ABSL3) facility. On day 31–32 for primed macaques (two animals per group per day for technical reasons) or day 58 for primed/boosted macaques, animals were challenged intranasally and intratracheally with 6.3 $\log_{10}$ $TCID_{50}$ of SARS-CoV-2 USA-WA1/2020. Post-challenge samples were collected as described for the post-immunization sampling. On day 6 pc, animals were euthanized and six separate samples from individual lung lobes were collected, snap frozen in dry ice, and stored at −80 °C.

### Replication of MPV and derivatives in the upper and lower airways of macaques

MPV and MPV/S-2P shedding in the upper and lower airways of macaques was evaluated by dual-staining immunoplaque assay of NS and TL samples. Briefly, tenfold serial dilutions were incubated in duplicate wells on Vero cells under methyl-cellulose overlay. After 11-day incubation at 32 °C, monolayers were fixed using 80% methanol, MPV was detected using a rabbit hyperimmune MPV antiserum[4], and SARS-CoV-2 S-2P was detected using the human monoclonal antibody CR3022[37]. Anti-rabbit IRDye680 and anti-human IRDye800 infrared fluorophore-labeled secondary antibodies were used to visualize MPV and S using a LI-COR Odyssey Clx imager (LI-COR Biosciences). Plaques were visualized using ImageStudio Lite 5.2.5 (LI-COR Biosciences).

### Dual IgG and IgA ELISAs

Levels of serum and mucosal anti-SARS-CoV-2 S or anti-receptor binding domain (RBD) IgG and IgA elicited by MPV/S-2P were determined using a dual IgG/IgA or IgG/dimeric IgA ELISA. Serum, BAL, and NW samples were heated at 56 °C for 30 min to inactivate complement and reduce potential risk from any residual viruses. Black ninety-six−well plates (MaxiSorp, Thermo Fisher Scientific, cat #437111) were coated with 100 μl/well of SARS-CoV-2 S-2P or S-6P (1 μg/ml) or RBD (2 μg/ml)[35,38] in 50 mM carbonate coating buffer, and incubated overnight at 4 °C. Plates were washed three times with 250 μl washing buffer [PBS with 0.1% IGEPAL CA-630] and blocked with 250 μl DPBS containing 5% dry milk (W/V). Samples were serially diluted in sample dilution buffer (PBS + 5% dry milk + 0.2% IGEPAL CA-630) and transferred to antigen-coated assay plates in duplicate. After 1-h incubation, plates were washed as above. 100 μl per well of secondary antibodies [goat anti-monkey IgG(H + L)-HRP (Thermo Fisher, cat# PA1-84631, 1:10,000) and goat anti-monkey IgA-alpha chain-specific biotin (Alpha Diagnostic International, cat# 70049, 1:5000) or mouse anti-rhesus J chain-biotin (NHPRR, cat# PR-3316, 1:5000)] in dilution buffer was added, and plates were incubated for 1 h. Plates were washed and 100 μl per well of diluted Streptavidin-Europium (PerkinElmer, cat# 1244-360), diluted 1:2000 in PBS + 0.2% IGEPAL CA-630, was added. Plates were incubated for 1 h and washed. Then, 50 μl of Pierce ECL (Thermo Fisher, cat# 32106) per well was added and plates were read on the Synergy Neo2 (BioTek) plate reader to collect IgG luminescence

data. Plates were washed, and 100 μl per well of Enhancement Solution (PerkinElmer: 4001-0010) was added. Plates were read again using a program for time-resolved fluorescence (TRF; excitation 360/40; emission 620/40) to collect IgA data. Data were processed as follows: (1) the average reading was calculated from duplicate wells, (2), the average reading from blank samples was subtracted, (3) the cut-off value was set to the blank average plus three standard deviations. The IgG and IgA titers were determined by interpolating the sigmoid standard curve generated on GraphPad Prism version 9.0.

### Serum antibody avidity assay

The avidity of serum antibodies to S-6P was determined by modification of the dual IgG and IgA ELISA protocol. After incubation with diluted sera and before addition of secondary antibody mixtures, one set of plates was incubated with 100 μl PBS, while the other set of duplicate plates were treated with 100 μl 1.0 M sodium thiocyanate solution (NaSCN; Sigma-Aldrich) for 15 min to strip low-affinity S-6P binding antibodies. Plates were washed prior to addition of secondary antibodies. Anti-S IgG and IgA titers of each serum sample in presence or absence of NaSCN were calculated. The avidity index (AI) was calculated as the ratio of the NaSCN-treated IgG or IgA titers to the PBS-treated serum IgG or IgA titers. A maximum avidity index of 1 corresponds to no loss of antibody binding to S following NaSCN treatment.

### Serum neutralization assay of SARS-CoV-2 and MPV

Neutralization assays to determine serum antibody titers against SARS-CoV-2 WA1/2020, B.1.1.7, or B.1.351 of immunized macaques were done in a BSL3 laboratory. Sera were heat-inactivated at 56 °C for 30 min and two-fold serially diluted in Opti-MEM prior to incubation 1:1 with 100 $TCID_{50}$ of SARS-CoV-2 for 1 h at 37 °C. Mixtures were added to quadruplicate wells of Vero E6 cells in 96-well plates and incubated for 4 days. The 50% neutralizing dose ($ND_{50}$) was the highest dilution of serum that prevented cytopathic effect in 50% of the wells. Serum neutralizing antibody titers against the MPV vector were also quantified on Vero cells using 60% plaque reduction neutralization tests ($PRNT_{60}$).

### ACE2 binding inhibition assays

As a complement to the neutralization assay, we evaluated the ability of heat-inactivated sera and BAL fluid to inhibit binding of ACE2 to SARS-CoV-2 spike proteins (Meso Scale Diagnostics, cat# K15586U, K15609U, K15671U). Sera were diluted 1:20 and BAL samples were diluted 1:2. Each sample was evaluated in duplicate, and plates were prepared and analyzed following the manufacturer's instructions[8]. Briefly, plates in which each well was coated with 10 different spike proteins were blocked for 1 h using MSD blocker A buffer, followed by a wash with MSD washing buffer. Diluted test samples were added, and plates were further incubated for 2 h on a plate shaker. Sulfo-Tag labeled soluble ACE2 was added and after 1 h incubation, plates were washed. The MSD GOLD electrochemiluminescence read buffer B was added, and chemiluminescence of bound ACE2-Sulfo-Tag was acquired on a MESO Quickplex SQ 120MM reader (Meso Scale). Data were analyzed using Methodical Mind 1.0.38 (Meso Scale). The ACE2 binding inhibition is calculated as percent inhibition relative to no-sample controls.

### Identification of S-specific B cells in the blood of immunized macaques

To characterize S-specific B cells in the blood of immunized macaques, single-cell suspensions of PBMCs were plated in 96 well plates at ≤3.5 × 10[6] cells/ml in 200 μl PBS + 1% FBS. Cells were centrifuged at 544 × g for 5 min at 4 °C and incubated first with 0.3 μg each of purified SARS-CoV-2 S-2P and RBD probes, tagged with BV605 and BV421, respectively (see Supplementary Methods 1), for 30 min at room temperature (RT). To further characterize the S-specific B cells, PBMCs

were stained with a 50 µl cocktail of fluorochrome-labeled antibodies diluted in Wash Buffer for 20 min at 4 °C. The antibodies for surface staining were: IgD (FITC, Southern Biotech cat# 2030-02, at 1:50), CD3 (BB700, clone SP34-2, BD Biosciences cat# 566518, at 1:50), CD16 (BB700, clone 3G8, BD Biosciences cat# 746199, at 1:50), CD14 (BB700, clone M5E2, BD Biosciences cat# 745790, at 1:50), IgM (BV510, clone G20-127, BD Biosciences cat# 563113, at 1:50), CD27 (BV711, clone O323, Biolegend cat# 302834, at 1:20), CD20 (BV785, clone 2H7, Biolegend cat# 302356, at 1:20), CD21 (BUV395, clone B-ly4, BD Biosciences cat# 740288, at 1:20), IgG (BUV737, clone G18-145, BD Biosciences cat# 612819, at 1:20), CD138 (APC, clone MI15, Biolegend cat# 356506, at 1:20), Fixable Viability Dye eFlour780 (Thermo Fisher cat# 65-0865-18, at 1:50,000), CD38 (PE, clone OKT10, Caprico Biotechnologies, cat# 100826, at 1:50), IgA (Texas Red, Southern Biotech cat# 2050-07, at 1:10), CD19 (PE-Cy5, clone J3-119, Beckman cat# IM2643U, at 1:20), and CD95 (PE-Cy7, clone DX2, Biolegend cat# 305622, at 1:20). Cells were fixed overnight at 4 °C with the eBioscience Intracellular Fixation & Permeabilization Buffer Set (Thermo Fisher Scientific, cat# 88-8824-00) and analyzed on a BD FACS Symphony A5. Data were analyzed with FlowJo version 10.

### Identification and characterization of S-specific CD4$^+$ and CD8$^+$ T-cells in the blood and airways of immunized macaques

Blood was collected in EDTA tubes, diluted 1:1 with PBS, and PBMCs were isolated by density gradient centrifugation[8]. Briefly, 15 ml of Ficoll-Paque density gradient (GE Healthcare) was added to Leucosep PBMC isolation tubes (Greiner bio-one) and centrifuged at $1000 \times g$ for 1 min at 22 °C to collect Ficoll below the separation filter. The blood and PBS mixture was added to the Leucosep tubes with Ficoll-Paque and centrifuged at $863 \times g$ for 10 min at 22 °C. The upper layer was poured into a 50 ml conical tube and brought to 50 ml with PBS, and then centrifuged at $544 \times g$ for 5 min at 4 °C. The cell pellet was resuspended at $2 \times 10^7$ cells/ml, and aliquots of PBMCs were stored in 90% FBS and 10% DMSO in liquid nitrogen. BAL were first filtered through 100 µm filters (Corning, cat# 431752), and cells were collected by centrifugation for 5 min at $544 \times g$ at 10 °C. BAL fluid was separated from the cell pellet, aliquoted, snap frozen in dry ice and stored at −80 °C for further use. BAL cells were used fresh for evaluation of the S-specific CD4$^+$ and CD8$^+$ T-cell response in the lower airways.

For analysis of T-cells, thawed PBMCs that were rested overnight or freshly collected BAL cells were plated at $1 \times 10^7$ cells/ml in 200 µl in 96 well plates in X-VIVO 15 media supplemented with 10% FBS, 1000x Brefeldin (Thermo Fisher cat# 00-4506-51) and 1000x Monensin (Thermo Fisher cat# 00-4505-51) diluted 1:1000, CD107a (AF647, clone H4A3, Biolegend, cat# 328612) at 1:50, CD107b (AF647, clone H4B4, Biolegend, cat# 354312) at 1:50, and stimulated with the indicated peptide pools at 1 µg/ml for 6 h at 37 °C with 5% CO$_2$[8]. Spike peptide pools consisted of Peptivator SARS-CoV-2 Prot_S1 (Miltenyi cat# 130-127-048), Peptivator SARS-CoV-2 Prot_S+ (Miltenyi cat# 130-127-312), and Peptivator SARS-CoV-2 Prot_S (Miltenyi cat# 130-127-953) covering the whole spike protein. Nucleocapsid peptide pool consisted of Peptivator SARS-CoV-2 Prot_N (Miltenyi cat# 130-126-699). After stimulation, cells were centrifuged at $515 \times g$ for 5 min at 4 °C and washed once with PBS with 1% FBS and 0.05% sodium azide. Cells were resuspended in 50 µl extracellular stains diluted in PBS with 1% FBS and 0.05% sodium azide, and stained for 20 min at 4 °C using a panel of antibodies including CD69 (FITC, clone FN50, Biolegend cat# 310903, at 1:25), CD8a (eFluor 506, clone RPAT8, Thermo Fisher, cat# 69-0088-42, at 1:250), CD4 (BUV496, clone SK3, BD Biosciences, cat# 612937, at 1:13), CD95 (BUV737, clone DX2, BD Biosciences, cat# 612790, at 1:80), CD3 (BUV805, clone SP34-2, BD Biosciences cat# 566518, at 1:80), viability Dye eFluor780 (Thermo Fisher, cat# 65- 0865-18, at 1:50,000), CD103 (PE, clone B-Ly7, eBioscience, cat# 12-1038-42, at 1:25), CD28 (PE/Dazzle 594, clone CD28.2, Biolegend, cat# 302942, at 1:50). After extracellular staining, cells were washed twice with PBS with 1% FBS

and 0.05% sodium azide and fixed with eBioscience Intracellular Fixation & Permeabilization Buffer Set (Thermo Fisher Scientific, catalog no. 88-8824-00) for 16 h at 4°C. After fixation, cells were centrifuged at $974 \times g$ for 5 min at 4°C without brake and resuspended in 50 µl intracellular stains diluted in eBioscience permeabilization buffer, and stained for 30 min at 4°C using a panel of antibodies for intracellular staining[8], including granzyme B (BV421, clone GB11, BD Biosciences, cat# 563389, at 1:250), IL-2 (BV605, MQ-17H12, Biolegend, cat# 500332, at 1:20), IFNγ (BV711, clone 4S.B3, Biolegend, cat# 502540, at 1:50), IL-17 (BV785, clone BL168, Biolegend, cat# 512338, at 1:50), TNFα (BUV395, clone Mab11, BD Biosciences cat# 563996, at 1:80), Ki-67 (PE-Cy7, clone B56, BD Biosciences, cat# 561283, at 1:50), Foxp3 (AF700, clone PCH101, Thermo Fisher, cat# 56-4776-41, at 1:20). After staining, cells were washed with 2x eBioscience permeabilization buffer 2x and resuspended in PBS supplemented with 1% FBS and 0.05% sodium azide for flow cytometry analysis on a BD FACSymphony A5 (BD Biosciences). Data were analyzed using FlowJo version 10.

### Quantification of SARS-CoV-2 genomic N and subgenomic E by RT-qPCR

Viral RNA from NS and BAL was extracted using the QIAamp Viral RNA Mini Kit (Qiagen)[8]. Homogenized lung tissues were mixed with Trizol LS Reagent (Thermo Fisher) and RNA was extracted using Phasemaker (Thermo Fisher) and the PureLink RNA Mini Kit (Thermo Fisher, cat #12183018A) according to the manufacturer's instructions. SARS-CoV-2 genomic N RNA and subgenomic E mRNA were quantified in triplicate using the TaqMan RNA-to-Ct 1-Step Kit (Thermo Fisher, cat #4392938)[8] on a QuantStudio 7 Pro real-time PCR system (Thermo Fisher; sgE assay: forward primer: 5′-CGATCTCTTGTAGATCGTTCT C-3′, reverse primer: 5′-ATATTGCAGCAGTACGCACACA-3′, probe: 5′-ACACTAGCCATCCTTACTGCGCTTCG-3′; gN assay: forward primer: 5′-GACCCCAAAATCAGCGAAAT-3′, reverse primer: 5′-TCTGGTTACTGC-CAGTTGAATCTG-3′, probe: 5′-ACCCCGCATTACGTTTGGTGGAC C-3′[9,39,40]). Copy numbers were determined using standard curves generated using serially diluted pcDNA3.1 plasmids encoding gN or sgE sequences (QuantStudio 6/7 Pro touchscreen instrument operating software; Standard Curve application on https://apps.thermofisher.com).

### Statistical analysis

Data sets were analyzed for significance using one-way ANOVA with Tukey post-test or two-way ANOVA with Sidak's post-test on GraphPad Prism version 9.5. A $\log_{10}$ transformation was applied to data sets when necessary to obtain comparable standard deviation among groups of values. Data sets were only considered significantly different at $p \leq 0.05$.

### Reporting summary

Further information on research design is available in the Nature Portfolio Reporting Summary linked to this article.

## Data availability

The experimental data generated in this study are provided in the main text or in the Supplementary Information/Source Data File. Nucleotide sequences cited in this manuscript include: GenBank MN985325; GISAID accession ID: EPI_ISL_404895 [https://www.epicov.org/epi3/frontend#3f2c48] (SARS-CoV-2 USA-WA1/2020). GISAID: EPI_ISL_751801 [https://www.epicov.org/epi3/frontend#42f6ed] (SARS-CoV-2 USA/CA_CDC_5574/2020 isolate (lineage B.1.1.7)). GISAID: EPI_ISL_890360 [https://www.epicov.org/epi3/frontend#2a5858] (SARS-CoV-2 USA/MD-HP01542/2021 isolate (lineage B.1.351). GenBank AY729016 [https://www.ncbi.nlm.nih.gov/nuccore/58610194] (Murine pneumonia virus). GenBank MN908947 (SARS-CoV-2, isolate Wuhan Hu-1). Source data are provided with this paper.

## Materials availability

Materials described in the manuscript are available upon request under a material transfer agreement with the NIAID. Requests should be addressed to U.J.B.

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

## Acknowledgements

We thank the NIAID Comparative Medicine Branch staff for animal study support, Melanie Cohen for help with the Luminex assay, Tammy Krogmann and Wei Bu for help with the ADCP assay, and Peter L. Collins for helpful discussions and comments on the manuscript. This research was supported by the Intramural Research Programs of the Division of Intramural Research (Project numbers ZIA AI001298-01; U.J.B; ZIA AI001258-04; J.I.C.; and 1ZIAAI001294-04, D.L.B.) and the Vaccine Research Center, NIAID, NIH (Project number ZIC AI005111-14; P.D.K.).

## Author contributions

Conceptualization: C.L.N. and U.J.B. Methodology: J.A.K., C.E.N., X.L., H.S.P., Y.M., C.L., R.H., R.M., K.D., J.I.C., D.L.B., C.L.N., and U.J.B.. Investigation: J.A.K., C.E.N., X.L., H.S.P., Y.M., C.L., C.S., L.R.H.A., R.H., I.N.M., T.W.K., R.M., A.W., L.Y., K.D., H.N., J.K., N.L.G., L.E.V., and C.L.N. Data analysis and visualization: J.A.K., C.E.N., X.L., H.S.P., Y.M., C.L., C.S., R.H., I.N.M., T.W.K., R.M., A.W., L.Y., K.D., H.N., J.K., J.I.C., N.L.G., L.E.V., D.L.B., U.J.B., and C.L.N. Writing—original draft: J.A.K., C.L.N., and U.J.B. Writing—review and editing: J.A.K., C.E.N., X.L., H.S.P., Y.M., C.L., C.S., L.R.H.A., R.H., I.N.M., T.W.K., R.M., A.W., L.Y., I.T.T., P.D.K., K.D., H.N., J.K., J.I.C., R.F.J., N.L.G., L.E.V., S.M., D.L.B., C.L.N., and U.J.B.

## Funding

## Competing interests

J.A.K., C.L., S.M., C.L.N. and U.J.B. are inventors on the provisional patent application number 63/502,829 entitled "Recombinant murine pneumonia virus expressing severe acute respiratory syndrome coronavirus 2 (SARS-COV-2) spike protein" filed by the United States, Department of Health and Human Services. The remaining authors declare no competing interests.

## Additional information

[1]RNA Viruses Section, Laboratory of Infectious Diseases, National Institute of Allergy and Infectious Diseases, National Institutes of Health, Bethesda, MD, USA. [2]T-Lymphocyte Biology Section, Laboratory of Parasitic Diseases, National Institute of Allergy and Infectious Diseases, National Institutes of Health, Bethesda, MD, USA. [3]Experimental Primate Virology Section, Comparative Medicine Branch, National Institute of Allergy and Infectious Diseases, National Institutes of Health, Poolesville, MD, USA. [4]Comparative Medicine Branch, National Institute of Allergy and Infectious Diseases, National Institutes of Health, Bethesda, MD, USA. [5]Division of Pathology, Emory National Primate Research Center, Emory University, Atlanta, GA, USA. [6]Division of Assurances, Office of Laboratory Animal Welfare, National Institutes of Health, Bethesda, MD, USA. [7]Emory National Primate Research Center, Environmental Health and Safety Office, Emory University, Atlanta, GA, USA. [8]Tuberculosis Imaging Program, Division of Intramural Research, National Institute of Allergy and Infectious Diseases, National Institutes of Health, Bethesda, MD, USA. [9]Vaccine Research Center, National Institute of Allergy and Infectious Diseases, National Institutes of Health, Bethesda, MD, USA. [10]Medical Virology Section, Laboratory of Infectious Diseases, National Institute of Allergy and Infectious Diseases, National Institutes of Health, Bethesda, MD, USA. [11]SARS-CoV-2 Virology Core, Laboratory of Viral Diseases, National Institute of Allergy and Infectious Diseases, National Institutes of Health, Bethesda, MD, USA. [12]These authors contributed equally: Christine E. Nelson, Xueqiao Liu. [13]These authors jointly supervised this work: Ursula J. Buchholz, Cyril Le Nouën. ✉e-mail: ubuchholz@niaid.nih.gov; lenouenc@niaid.nih.gov

