## [Peer Review File · Nature Communications]

Mucosal prime-boost immunization with live murine pneumonia virus-vectored SARS-CoV-2 vaccine is protective in macaquesReviewers' Comments:

Reviewer #1:

Remarks to the Author:

In this study Kaiser et al. investigated the immunogenicity and efficacy of a live murine pneumonia virus (MPV) vectored SARS-CoV2 vaccine administered intranasally (IN) plus intratracheally (IT) either as a single dose or two doses in rhesus macaques. They performed a series of immune measurements to define the B and T cell responses in the upper and lower airways, and in blood. These include the binding antibody responses to Spike (S) and RBD proteins, memory B cells specific to S and RBD, ACE2 inhibition, live virus neutralizing Ab response, and S-specific CD4 and CD8 T cell responses in blood and BAL. To determine efficacy, they challenged animals IN and IT with SRAS-CoV2 WA-1 strain. The authors performed a thorough and exhaustive analysis of B and T cell responses. Their results showed that MPV mucosal vaccination induces strong antibody and T cell responses including in the lower airway, and provides protection in the lower airway. Discouragingly, mucosal vaccination failed to protect the upper airway.

The main concern I have with this manuscript is the significance of the findings. Multiple studies previously showed protection in the lower airway from SARS-CoV2 following intranasal/intratracheal vaccination. In fact, protection in the lower airway was not different from that has been reported by IM vaccinations. In addition, the mucosal vaccination did not provide protection in the upper airway, which is an unmet goal currently. The authors used WA-1 virus for challenge, which is not relevant for the current circulating virus.

Reviewer #2:

Remarks to the Author:

This work evaluated an MPV-vectored SARS-CoV-2 live-attenuated vaccine in rhesus macaques. This live-attenuated vaccine was designed to express the SARS-CoV-2 Spike protein in a non-human respiratory virus MPV, and directly delivered to macaques by intranasal/intratracheal route. The authors assessed the mucosal immunity and protective efficacy after prime-boost immunization, and they found a single dose of MPV/S-2P efficiently induced systemic and mucosal immunity to protect the macaques from WT SARS-CoV-2 infection. At the same time, the boost strongly increased more multifunctional antibody responses in blood and airways, and the boost also induced restimulation and proliferation of spike-specific T cells. Overall, the study was well-designed, and the manuscript was well-written.

Comments:

A very similar strategy has been reported using B/HPIV3 as another respiratory viral vector, which was developed as a pediatric vaccine candidate and has been proven safe for infants and young children. That one could also induce very efficient/comparable mucosal immunity and antibody response to protect the animals from challenge with a single dose, what are the advantages of MPV/S-2P compared with B/HPIV3/S-6P?

The induced neutralizing antibody against previous isolates has been shown in the current version.

However, the hACE2 affinity can not fully reflect the antibody neutralization, for instance, in the boost samples, WA1/2020 and B.1.1.7 show ~ 2.0 (Log10) ND50 in Fig. 3D with 100% inhibition of hACE2 binding, while the ACE2 binding was 100% inhibited in B.1.351 with the ND50 of ~ 1.5 (Log 10).

Therefore, it will be interesting to perform lentivirus pseudotype virus neutralization assays to evaluate the more SARS-CoV-2 variants with different antibody escape abilities.

In the challenge experiment, I would like to use a most recent or more pathogenic instead of using the original SARS-CoV-2 strain. Otherwise, the challenge protection will only prove the concept.

Reply to the reviewers' comments

Thank you for the insightful comments on our manuscript entitled “Mucosal prime-boost immunization with live murine pneumonia virus-vectored SARS-CoV-2 vaccine is protective in macaques”. We were happy that the comments were overall positive.

We have included our point-by-point replies to each comment below, and we have revised the manuscript in response to the comments. We hope that our revisions have improved the manuscript.

Reviewer #1:

In this study Kaiser et al. investigated the immunogenicity and efficacy of a live murine pneumonia virus (MPV) vectored SARS-CoV2 vaccine administered intranasally (IN) plus intratracheally (IT) either as a single dose or two doses in rhesus macaques. They performed a series of immune measurements to define the B and T cell responses in the upper and lower airways, and in blood. These include the binding antibody responses to Spike (S) and RBD proteins, memory B cells specific to S and RBD, ACE2 inhibition, live virus neutralizing Ab response, and S-specific CD4 and CD8 T cell responses in blood and BAL. To determine efficacy, they challenged animals IN and IT with SRAS-CoV2 WA-1 strain. The authors performed a thorough and exhaustive analysis of B and T cell responses. Their results showed that MPV mucosal vaccination induces strong antibody and T cell responses including in the lower airway, and provides protection in the lower airway. Discouragingly, mucosal vaccination failed to protect the upper airway.

We thank the reviewer for the positive comments.

However, the last sentence of the summary (“Discouragingly, mucosal vaccination failed to protect the upper airway”) does not capture the key results of our study. We would like to clarify: our results show that two doses of MPV/S-2P, administered by the respiratory mucosal route, conferred complete protection against high-dose SARS-CoV-2 challenge in the upper and lower airways of macaques; one dose induced close-to-complete protection. Based on its ability to protect the upper as well as the lower airways, the MPV/S-2P vaccine candidate might fill an unmet public-health need.

We have clarified in the revised manuscript that the qRT-PCR specific for subgenomic E (sgE) RNA detects SARS-CoV-2 mRNA/RNA synthesis, indicative of SARS-CoV-2 infection and replication. sgE mRNA is transcribed only in infected cells, not packaged in virions, and serves as a marker for infection and active SARS-CoV-2 challenge virus transcription/replication (Results, lines 281-284), while the assay specific for SARS-CoV-2 N genomic RNA (gN) detects the mere presence of challenge virus, including input virus, irrespective of replication (lines 293, 294). Positive signals in the gN assay alone in animals challenged with high doses of SARS-CoV-2 do not indicate lack of protection against challenge. To highlight and clarify the main message on protective efficacy against SARS-CoV-2 challenge in the upper (and lower) airways, we have revised figure 7, keeping the sgE taqman results in the main figure, and moving the accessory gN taqman results to the new supplementary figure S6.

The high levels of gN RNA (confirming the high dose of challenge virus received) in conjunction with the negative results of the sgE assay show that the level of protection against challenge virus replication induced by two doses of MPV/S-2P was actually impressive.

The main concern I have with this manuscript is the significance of the findings. Multiple studies previously showed protection in the lower airway from SARS-CoV2 following intranasal/intratracheal vaccination. In fact, protection in the lower airway was not different from that has been reported by IM vaccinations. In addition, the mucosal vaccination did not provide protection in the upper airway, which is an unmet goal currently.

As described in our response to the first comment above, our results show efficacy of MPV/S-2P. Two doses of MPV/S-2P, administered by the respiratory mucosal route, induced complete protection against high-dose SARS-CoV-2 challenge in the upper and lower airways of macaques; one dose induced close-to-complete protection.

In this study, we evaluated protection against challenge virus replication by using a qRT-PCR assay specific for SARS-CoV-2 subgenomic E mRNA. This is the most sensitive assay to detect newly synthesized sgE mRNA, indicative of SARS-CoV-2 infection and challenge virus replication (lines 281-284; Fig. 7). For completeness, we also evaluated the loads of genomic N RNA, revealing the presence of high levels of input challenge virus (lines 293-300, results originally presented in the right panels of Figure 7AC; now moved to supplementary Figure S6).

We have revised the manuscript to clarify that based on undetectable subgenomic E (sgE) RNA (i.e., undetectable SARS-CoV-2 transcription/replication, Fig. 7), in conjunction with high levels of gN RNA (i.e., high levels of input challenge virus), two doses of MPV/S-2P induced complete protection against challenge virus replication in the upper and lower airways following high-dose SARS-CoV-2 challenge (Results, lines 290-292 and 300-305). To clarify our main message on protective efficacy against SARS-CoV-2 challenge, we have revised figure 7, keeping the sgE taqman results in the main figures, and presenting the accessory gN taqman results in the new supplementary figure S6. We also have added background information on the RT-qPCR assays in the legends of both figures (Lines 933-934 and 1143).

We agree that IM vaccination with mRNA-based vaccines similarly provides full protection in lower airways. However, while injectable COVID vaccines do not effectively prevent SARS-CoV-2 infection and replication in the upper airways, we found that mucosal immunization with two doses of MPV/S-2P provided complete protection against SARS-CoV-2 challenge virus replication in the upper airways, in addition to protecting the lower airways of macaques. We have revised the discussion to highlight this point (lines 413-418: “While strong protection in the lower airways has also been shown for injectable vaccines, including mRNA and adenovirus vectored vaccines (Corbett et al, PMID 32722908 Mercado et al., PMID 32731257, van Doremalen, PMID 32731258), protection by MPV/S-2P against SARS-CoV-2 challenge virus replication in the upper respiratory tract was more robust compared to injectable vaccines and predicts that MPV/S-2P immunization may restrict SARS-CoV-2 transmission in vaccinated individuals”. Based on these results, MPV/S-2P was selected for evaluation in a Phase 1 study as a second-generation COVID vaccine for intranasal immunization.

As discussed in our manuscript, mucosal immunization with a live-attenuated vector such as MPV/S-2P offers several advantages over IM administered vaccines. Unlike injectable vaccines, mucosal immunization with MPV/S-2P induced spike-specific dimeric IgA in both upper and lower airways. These antibodies are typically highly neutralizing and while this has not been investigated in this study, should also reduce or prevent SARS-CoV-2 shedding and transmission. In addition, our mucosal antibody results suggest that MPV/S-2P induced spike-specific memory B cell locally that could be beneficial for long-term protection in the airways (lines 353-358, 375-379 of the discussion).

In addition, unlike injectable vaccines (that are not known to induce S-specific T cells in the airways; Gagne et al., 2022, Cell; PMID:35447072), MPV/S-2P induced robust S-specific CD4⁺ and CD8⁺ T cells in the airways that transitioned to tissue resident memory T cells. Previous studies suggested that tissue resident memory T cells typically are long-lived in the airways and provide long-term immunity to respiratory infections (references 12-15; lines 380-383, 390-394 of the discussion).

As protection induced by IM administered vaccines wanes relatively rapidly (CDC, MMWR, May 26, 2023), mucosal administration of a live attenuated viral vector such as MPV/S-2P may help to

increase the duration of protection and prevent or reduce transmission. We hope that these points support the significance of our findings.

The authors used WA-1 virus for challenge, which is not relevant for the current circulating virus.

We understand the reviewer's concern that WA-1 is not the currently circulating strain of SARS-CoV-2. This study served as a proof of concept that MPV/S2-P could be used as a live-attenuated vector vaccine to protect against a matched SARS-CoV-2 challenge virus. Our results show that MPV is an attenuated vector that is safe, immunogenic, and protective in macaques. A clinical trial is in preparation to evaluate the safety and immunogenicity of MPV/S-2P in healthy adults. We are currently generating clinical trial material of MPV expressing spike of a currently circulating variant to prepare for clinical studies beyond Phase 1. We have included this information in lines 432-433 of the modified manuscript.

Reviewer #2:

This work evaluated an MPV-vectored SARS-CoV-2 live-attenuated vaccine in rhesus macaques. This live-attenuated vaccine was designed to express the SARS-CoV-2 Spike protein in a non-human respiratory virus MPV, and directly delivered to macaques by intranasal/intratracheal route. The authors assessed the mucosal immunity and protective efficacy after prime-boost immunization, and they found a single dose of MPV/S-2P efficiently induced systemic and mucosal immunity to protect the macaques from WT SARS-CoV-2 infection. At the same time, the boost strongly increased more multifunctional antibody responses in blood and airways, and the boost also induced restimulation and proliferation of spike-specific T cells. Overall, the study was well-designed, and the manuscript was well-written.

We thank the reviewer for the positive comments about our study.

Comments:

A very similar strategy has been reported using B/HPIV3 as another respiratory viral vector, which was developed as a pediatric vaccine candidate and has been proven safe for infants and young children. That one could also induce very efficient/comparable mucosal immunity and antibody response to protect the animals from challenge with a single dose, what are the advantages of MPV/S-2P compared with B/HPIV3/S-6P?

We previously reported the evaluation of the chimeric bovine/human parainfluenza type 3 (B/HPIV3) vector expressing SARS-CoV-2 spike in rhesus macaques (Le Nouen et al., 2022 Cell; PMID: 36423629). B/HPIV3/S-6P is designed to provide dual protection against SARS-CoV-2 and HPIV3. However, B/HPIV3 vectors rely on HPIV3 glycoproteins to infect. Since older children and adults typically have pre-existing immunity to HPIV3, the use of B/HPIV3/S-6P will be likely limited to the pediatric population. Since there is no evidence of pre-existing immunity against MPV in humans, an MPV-based vector would not be limited to a specific age group.

We have revised the discussion to clarify; "B/HPIV3/S-6P was primarily designed as a pediatric vaccine to protect against HPIV3 and COVID; with MPV/S-2P, a live intranasal vaccine candidate without pre-existing immunity in humans is available for clinical evaluation in adults." (lines 428-430). Also, MPV/S-2P vector-specific immunity induced by the first dose in macaques unexpectedly did not suppress the booster immune response to the S antigen following the second dose, suggesting that the MPV platform might be effective in homologous prime-boost immunizations for diverse age groups (Discussion, lines 423-425).

The induced neutralizing antibody against previous isolates has been shown in the current version. However, the hACE2 affinity can not fully reflect the antibody neutralization, for instance, in the boost samples, WA1/2020 and B.1.1.7 show ~2.0 (Log10) ND50 in Fig. 3D with 100% inhibition of hACE2 binding, while the ACE2 binding was 100% inhibited in B.1.351 with the ND50 of ~1.5 (Log

10). Therefore, it will be interesting to perform lentivirus pseudotype virus neutralization assays to evaluate the more SARS-CoV-2 variants with different antibody escape abilities.

In the present study, we have evaluated the breadth of the mucosal and serum antibody response to SARS-CoV-2 variants by ACE2 inhibition assay. In addition, we evaluated the serum neutralizing antibody titers against the vaccine-matched SARS-CoV-2 WA1/2020 strain as well as two early variants of concern (Alpha/B.1.1.7 and Beta/B.1.351).

ACE2 binding inhibition assays are highly sensitive multiplex assay with a big dynamic range that require lower serum volumes than live or pseudotype virus neutralization assays. In a previous study (Le Nouen et al., 2022 Cell; PMID: 36423629), we found that pseudotype virus neutralization assays and ACE2 binding inhibition assays provided overall comparable results, rather than providing additional insights. However, the ACE2 binding inhibition assays were more sensitive than pseudotype neutralization assays; using the ACE2 binding inhibition assays, we were more successful in evaluating the breadth of S specific antibodies in mucosal samples from rhesus macaques. Based on these previous results, we opted for the analysis of the breadth of S-specific serum and mucosal antibodies by the ACE2 binding inhibition assay rather than by pseudotype neutralization assay in our present study. We have added a note to indicate this (results, line 124, addition underlined: This assay evaluated the ability of antibodies to inhibit binding of soluble, tagged ACE2 to purified S protein ... and, in a past study, was more sensitive than SARS-CoV-2 pseudotype neutralization assays (Le Nouen et al., 2022 Cell; PMID: 36423629). In addition, we performed live virus neutralization assays to detect serum antibodies against WA1/2020 and B.1.1.7 variants.

In the challenge experiment, I would like to use a most recent or more pathogenic instead of using the original SARS-CoV-2 strain. Otherwise, the challenge protection will only prove the concept.

We understand the reviewer's concern that WA-1 is not the currently circulating strain of SARS-CoV-2. This study serves as a proof of concept that MPV could be used as a live attenuated vector vaccine against an antigenically matched isolate of SARS-CoV-2. Based on the safety and immunogenicity of MPV/S-2P in macaques, this vaccine candidate was selected for a clinical trial in healthy adults. MPV/S-2P is easily adaptable to new emerging variants, and MPV vaccine candidates expressing spike of the currently circulating variants are being prepared for clinical evaluation beyond Phase 1. We have added this point at the end of the discussion (lines 432-433).

Reviewers' Comments:

Reviewer #2:

Remarks to the Author:

The authors have addressed my concerns.

I want to clarify that when we performed the antibody neutralization assay, it included attachment, entry (internalization/fusion), uncoating and initial translation, which information can be provided by the pseudotype virus assay because they were in the same backbone. Actually, I would like to do the chimeric spike viruses in the same viral backbone, only including different spike sequences, but it is not allowed due to safety issues.

Reply to the reviewers' comments:

Reviewer #2 (Remarks to the Author):

The authors have addressed my concerns.

I want to clarify that when we performed the antibody neutralization assay, it included attachment, entry (internalization/fusion), uncoating and initial translation, which information can be provided by the pseudotype virus assay because they were in the same backbone. Actually, I would like to do the chimeric spike viruses in the same viral backbone, only including different spike sequences, but it is not allowed due to safety issues.

We thank Reviewer 2 for the positive feedback on our revisions, and for the clarification.